# DeFacto: Counterfactual Thinking with Images for Enforcing Evidence-Grounded and Faithful Reasoning

## Abstract

Recent advances in multimodal language models (MLLMs) have achieved remarkable progress in vision-language reasoning, especially with the emergence of "thinking with images," which integrates explicit visual steps into the reasoning process. While this paradigm strengthens image-based reasoning, a significant challenge remains: models may arrive at correct answers by relying on irrelevant or spurious regions, driven by prior knowledge or dataset biases. Even when the answer is correct, flawed reasoning indicates that the model has not truly understood the image, highlighting the critical importance of reasoning fidelity in multimodal tasks. To address this issue, we propose *DeFacto*, a counterfactual reasoning framework that jointly enforces accurate answering and faithful reasoning. A key component of our approach is the design of three complementary training paradigms: (i) positive, (ii) counterfactual, and (iii) random-masking. To enable these paradigms, we develop a pipeline that automatically localizes question-relevant evidence and constructs positive, counterfactual, and random variants, resulting in a dataset of about 100k images. Building on this framework, we train multimodal language models with GRPO-based reinforcement learning, where we design three complementary rewards to guide the model toward accurate answering and evidence-grounded reasoning. Experiments on diverse benchmarks demonstrate that *DeFacto* substantially improves both answer accuracy and reasoning faithfulness, establishing a stronger foundation for interpretable multimodal reasoning. The code and datasets will be released upon acceptance.

## 1 Introduction

Vision-language models (VLMs) Alayrac et al. (2022); Li et al. (2023a); Zhu et al. (2023); Liu et al. (2023; 2024a); Peng et al. (2023); Bai et al. (2023); Team et al. (2023); Chen et al. (2024c;b) have achieved remarkable progress in recent years, demonstrating strong capabilities across a wide range of multimodal tasks such as visual question answering, image captioning, and referring expression comprehension. By leveraging large-scale pretraining and cross-modal alignment, these models can generate fluent and semantically relevant outputs grounded in visual context. However, in complex scenarios that require multi-step reasoning or fine-grained perception, existing models often rely heavily on implicit language priors, producing plausible yet unfaithful responses that are weakly grounded in the actual image. Instead of genuinely learning to reason over visual content, these models often fall back on text-based chain-of-thought patterns, limiting their ability to handle cases where critical evidence must be directly perceived from the image.

Recent advances in "thinking with images" Microsoft (2024); OpenAI (2025) emphasize the integration of explicit visual steps into the reasoning process to enhance transparency and visual grounding. Early approaches employ supervised fine-tuning (SFT) Ouyang et al. (2022); Touvron et al. (2023); Liu et al. (2023); Dettmers et al. (2023), where models are trained in a chain-of-thought (CoT) Shao et al. (2024) manner to produce region-aware reasoning traces based on manually annotated visual steps. To reduce the annotation burden, subsequent works explore reinforcement learning strategies that allow models to autonomously develop visual interaction behaviors such as region cropping, attention shifting, or zooming Zheng et al. (2025); Cao et al. (2025); Liu et al. (2025b); Zhang et al. (2025b). Yet these approaches do not guarantee that the reasoning chains are faithful to the actual vi-

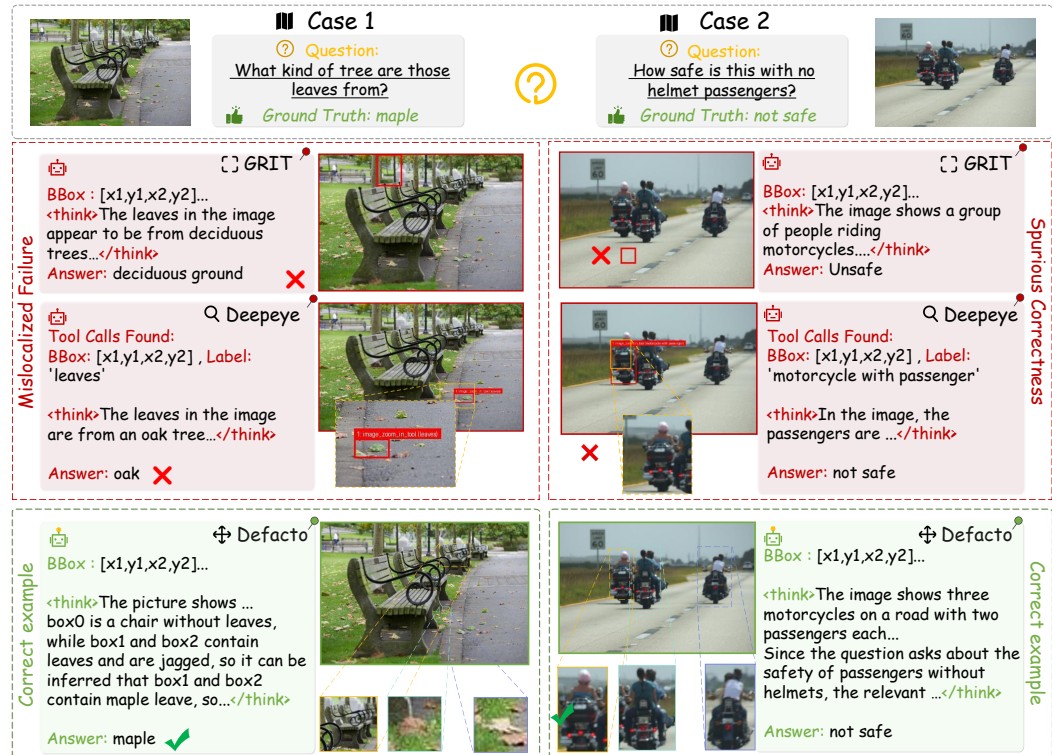

Figure 1: Qualitative examples of failure cases. Left: Mislocalized Failure (park scene). Right: Spurious Correctness (road scene).

sual evidence: since the model still has access to the entire image, it may either produce an incorrect answer by focusing on irrelevant regions or arrive at the correct answer even when the highlighted regions are unrelated. This issue is clearly illustrated in Fig. 1, where most existing models exhibit two characteristic error modes: *Mislocalized Failure*, in which the model selects irrelevant regions and consequently produces an incorrect answer, and *Spurious Correctness*, in which the answer happens to be correct even though the selected regions are unrelated to the reasoning process. In the park scene (left), GRIT mistakenly attends to the distant background, while Deepeyes zooms in on a faint and indistinct fallen leaf, both of which fail to capture the evidence and lead to an incorrect answer. In the road scene (right), GRIT fixates on the ground, and Deepeyes again zooms in on a helmeted rider, producing a reasoning path that contradicts the very premise of the question. These cases reveal a deeper problem: current approaches can still succeed superficially even when their reasoning is disconnected from the actual evidence. However, correct answers alone are not sufficient—the reasoning process itself must also be correct, since flawed reasoning often leads to erroneous predictions. Such superficial success leads to poor generalization on out-of-distribution inputs and undermines trustworthiness in downstream applications that demand evidence-based decisions. What is needed is a training paradigm that enforces both correct evidence selection and correct answering, ensuring that reasoning trajectories and final predictions are jointly faithful to the visual input.

Motivated by these failure cases, we introduce *DeFacto*, a counterfactual reasoning framework that aligns reasoning trajectories with visual evidence, ensuring predictions are both reliable and interpretable. The core idea is to employ three complementary training forms that jointly constrain the model's behavior: (i) positive supervision, (ii) counterfactual abstention, and (iii) random masking to strengthen evidence-grounded reasoning. In the positive case, the model is given the original image and trained to predict bounding boxes that cover the essential evidence together with the correct answer, receiving positive feedback only when both the evidence selection and the answer are correct. In the counterfactual case, the same question is paired with an image where the evidence regions $\mathcal{R}^+$ have been masked; since the necessary visual evidence is no longer available, the model is expected to abstain by outputting a designated token such as "unknown," while any concrete an-

swer is penalized. In the random-masking case, irrelevant regions $\mathcal{R}^-$ are masked independently of the question, preventing the model from exploiting superficial correlations between the presence of masks and abstention behavior. Training is performed with GRPO-based reinforcement learning, where the reward integrates three components: (i) Answer Correctness Reward, (ii) Format Consistency Reward, and (iii) Region Selection Coherence Reward. Through this design, DeFacto compels the model to produce reasoning that is logically grounded, answers that are accurate, and predictions where reasoning and outcomes remain consistent.

In practice, constructing counterfactual samples requires reliably identifying the question-relevant regions. To this end, we adopt a structured two-stage extraction pipeline. First, a multimodal language model (Qwen2.5-VL Bai et al. (2025)) parses the question and generates a set of key descriptors (e.g., "the red cup," "the text on the shirt"). Next, candidate regions in the image are obtained from a region proposal network (RPN) Ren et al. (2015) and an OCR module Islam et al. (2017). The OCR regions are further matched with textual descriptors to capture evidence critical for text-centric questions. For visual objects, the descriptors are fed into an open-vocabulary detector (DINO-X Ren et al. (2024)), which provides bounding boxes that serve as positive evidence regions. Finally, the remaining proposals from the RPN, after removing matched positives, are treated as irrelevant regions for counterfactual construction. Using this pipeline, we construct a counterfactual dataset about 100k images, ensuring that positive, counterfactual, and random-masking instances differ only in the availability of essential evidence while preserving unrelated context. Building on this dataset, the model is further optimized with GRPO-based reinforcement learning. This training paradigm enforces consistency between evidence selection and final predictions, ensuring that reasoning traces remain faithful to visual cues. As illustrated in Fig. 1, our method consistently grounds its reasoning in the correct regions (e.g., focusing on the three motorcycles on the road and their passengers), thereby unifying reasoning steps with faithful visual evidence.

Our main contributions are threefold:

(1) We propose a counterfactual "thinking with images" framework that aligns the reasoning process with essential visual evidence by jointly optimizing for answer correctness and region-level faithfulness via reinforcement learning. (2) We construct a new counterfactual dataset about 100k images using a language-guided algorithm that integrates open-vocabulary detection with targeted masking, ensuring that only question-relevant regions are removed while irrelevant context is preserved. (3) We demonstrate, through extensive experiments on diverse benchmarks, that our approach consistently improves both answer accuracy and visual grounding faithfulness over strong baselines.

## 2 RELATED WORK

**Structured Thinking with Images in Vision-Language Models.** The concept of "thinking with images" was initially highlighted in OpenAI o3 Achiam et al. (2023); OpenAI (2025) and later explored in works like COGCOM Qi et al. (2024) and GRIT Fan et al. (2025). Recent datasets also highlight the importance of evaluating visual reasoning beyond raw perception. For example, VisCoT Shao et al. (2024) provides visual-evidence for vqa, while datasets such as MSTI Chen et al. (2024d) emphasize structured visual understanding across detection, entity grounding, and answer generation. Existing approaches can be broadly categorized into two classes. The first category includes GRIT, which combines natural language and bounding boxes via reinforcement learning; REFOCUS Fu et al. (2025), which formulates visual editing as intermediate reasoning steps; COGCOM Qi et al. (2024), which models reasoning as visual manipulations such as cropping and OCR; and VisionReasoner Liu et al. (2025a), which unifies detection, segmentation, and counting under one framework. The second category emphasizes grounding quality. Fast-and-Slow Visual Agents Sun et al. (2024) model dual-system reasoning. DeepEyes Zheng et al. (2025) leverages reinforcement learning to train multimodal chains-of-thought and dynamically invoke zoom-in tools when visual evidence is ambiguous. MLLMs Know Where to Look Zhang et al. (2025a) improves small-object perception by applying inference-time cropping strategies to highlight fine details. Chain-of-Focus Zhang et al. (2025b) further adapts zoom-in operations through reinforcement learning, enabling multi-scale reasoning across cluttered scenes. Ground-R1 Cao et al. (2025) enhances faithfulness by introducing explicit reward signals that align reasoning outputs with grounded evidence. V* Wu & Xie (2024) formulates guided visual search as a core cognitive mechanism to explore high-resolution images efficiently. Visual-RFT Liu et al. (2025b) refines grounding via re-

inforcement fine-tuning. As a result, they often fail to ensure that reasoning trajectories remain consistent with the visual evidence, leaving open the need for a paradigm that jointly enforces faithful reasoning steps and accurate answers.

**Counterfactual Reasoning in Vision-Language Models.** Counterfactual reasoning in VLMs can be categorized into two types: counterfactual data generation and inference-based reasoning. The first enhances robustness by constructing or augmenting counterfactual samples to reduce bias and hallucination. For example, Learning Chain of Counterfactual Thought Zhang et al. (2020) disentangles factual knowledge from reasoning via CoBRa and CoCT datasets; C-VQA Zhang et al. (2024b) and CRIPP-VQA Patel et al. (2022) construct benchmarks for counterfactual VQA in static and video settings, respectively; Counterfactual Vision and Language Learning Abbasnejad et al. (2020) and Counterfactual Contrastive Learning Zhang et al. (2024c) generate counterfactuals through structural causal models and perturbation strategies, while CounterCurate Zhang et al. (2024a) improves compositional reasoning by augmenting training data with physically grounded examples and semantic counterfactuals using generative models. The second type focuses on inference with mechanisms such as Counterfactual-based Saliency Maps Wang et al. (2023) for contrastive visual explanation, DiG-IN Augustin et al. (2024) for diffusion-guided latent edits, and Counterfactual VQA Niu et al. (2021) for causal effect modeling. However, most existing approaches either treat counterfactuals as data augmentation without explicitly constraining the reasoning process, or apply them only at inference for explanation, leaving a gap in methods that can jointly enforce faithful reasoning steps and correct answers during training.

## 3 METHOD

In this section, we present the overall framework of DEFACTO, our counterfactual "thinking with images" approach. This section is organized into three parts: (1) the overall architecture and inference pipeline (Section 3.1); (2) the construction of counterfactual datasets via region masking and open-vocabulary filtering (Section 3.2); and (3) the reinforcement learning strategy with a tailored reward design that guides the model toward accurate answering, faithful reasoning, and their consistency (Section 3.3).

### 3.1 OVERALL FRAMEWORK

DEFACTO is a vision-language reasoning framework that enforces region-level faithfulness in multimodal question answering. It is designed to teach models not only *where to look* in the image but also *when to abstain* if the necessary evidence is absent. By combining structured prompting with counterfactual supervision, DEFACTO aligns the reasoning process with visual evidence rather than spurious correlations.

As illustrated in Figure 2, given a question and an image, the model is prompted to produce outputs in a structured format consisting of three fields. The `<bbox>` field contains one or more bounding boxes encoded as JSON objects of the form $\{\text{Position} : [x_1, y_1, x_2, y_2], \text{Confidence} : p\}$, the `<think>` field records a short rationale, and the `<answer>` field provides the final prediction. Multiple boxes can be returned when multiple evidence regions are required. If no valid evidence exists, the model outputs `unknown` in both the `<bbox>` and `<answer>` fields. This structured format ensures that every reasoning trajectory is explicitly tied to visual evidence through bounding boxes and aligned with the model's final answer. Training is based on three complementary supervision forms. In the positive case, evidence-bearing regions remain visible and the model is rewarded for selecting them and producing the correct answer. In the counterfactual case, these regions are masked, and the model is expected to abstain by outputting `unknown`. In the random-masking case, irrelevant regions are occluded to prevent shortcut learning from superficial mask patterns. Together, these three forms establish a consistent learning signal that requires both the reasoning path and the answer to be faithful to the underlying visual support.

### 3.2 COUNTERFACTUAL DATASET CONSTRUCTION

**Positive, Counterfactual, and Random Instances.** Let $\mathcal{R} = \{r_1, r_2, \ldots, r_n\}$ denote the set of candidate regions in an image $I$, obtained from a region proposal network (RPN) Ren et al. (2015) together with OCR to cover both object-level and text-bearing regions. Among them, $\mathcal{R}^+ \subseteq \mathcal{R}$

Figure 2: An overview of our counterfactual framework with three inputs: positive (full evidence), counterfactual (masked evidence), and random (masked irrelevant regions), guiding the model to answer correctly or abstain with "unknown."

represents the evidence regions that are critical to answering the question $q$, while $\mathcal{R}^- = \mathcal{R} \setminus \mathcal{R}^+$ denotes the remaining irrelevant regions. Based on these definitions, we construct three complementary training instances:

$$I_{\text{pos}} = (I, q, \mathcal{R}^+, y) \qquad I_{\text{cf}} = (I^{\setminus \mathcal{R}^+}, q, \mathcal{R}^-, y_{\text{cf}}) \qquad I_{\text{rand}} = (I^{\setminus \mathcal{R}^-}, q, \mathcal{R}^+, y), \qquad (1)$$

where $y$ is the ground-truth answer, $I_{\text{pos}}$ is the positive instance with evidence regions available, $I_{\text{cf}}$ is the counterfactual instance where evidence regions are masked and the abstention label $y_{\text{cf}}$ (e.g., "Unknown") is required, and $I_{\text{rand}}$ is the random-masking instance where irrelevant regions are occluded to prevent shortcut learning.

**Construction Process.** To automatically construct $I_{\text{pos}}$, $I_{\text{cf}}$, and $I_{\text{rand}}$ without manual annotations, we follow three steps:

(1) Descriptor extraction. Given an image $I$ and a question $q$, we employ a MLLM (Qwen2.5-VL Bai et al. (2025)) to extract a set of key descriptors:

$$\text{MLLM}(I, q) = \{d_1, d_2, \ldots, d_m\}, \qquad (2)$$

where each $d_i$ is a textual phrase (e.g., an object, attribute, or relation) that captures the visual concepts in $I$ explicitly mentioned or implied by $q$. As illustrated in Fig. 2 ("Automate Evidence Masking"), for the question "What does his shirt say?", the MLLM decomposes the query into descriptors such as "a man" and "man's shirt" as the critical evidence.

(2) Evidence localization. Let $\mathcal{R} = \{r_1, \ldots, r_n\}$ be the set of candidate image regions. We employ the open-vocabulary detector DINO-X Ren et al. (2024), which computes grounding scores $\text{OVD}(r, k)$ for each $r \in \mathcal{R}$ and $k \in \mathcal{K}(q)$. Based on these scores, the regions are partitioned into evidence and irrelevant sets:

$$\mathcal{R}^+ = \{r \in \mathcal{R} \mid \max_{k \in \mathcal{K}(q)} \text{OVD}(r, k) > \tau\}, \qquad \mathcal{R}^- = \mathcal{R} \setminus \mathcal{R}^+, \qquad (3)$$

where $\tau$ is a confidence threshold. In the street example, $\mathcal{R}^+$ corresponds to bounding boxes covering the signboard, while $\mathcal{R}^-$ contains all other regions.

(3) Instance generation. Once $\mathcal{R}^+$ and $\mathcal{R}^-$ are obtained, the positive, counterfactual, and random-masking instances are directly constructed as defined in Eq. 1.

For counterfactual dataset construction, we leverage a broad collection of visual question answering and document understanding benchmarks, including VQAv2 Goyal et al. (2017), OKVQA Marino et al. (2019), GQA Hudson & Manning (2019), ScienceQA Lu et al. (2022), VizWiz Gurari et al. (2018), TextVQA Singh et al. (2019), OCRVQA Mishra et al. (2019), AI2D Kembhavi et al. (2016), DocVQA Mathew et al. (2021), ChartQA Masry et al. (2022), InfoVQA Mathew et al. (2022), DeepForm Svetlichnaya (2020), Kleister KLC Stanisławek et al. (2021), WikiTableQuestions (WTQ) Pasupat & Liang (2015), TabFact Chen et al. (2019), and VisualMRC Tanaka et al. (2021). This diverse coverage ensures that counterfactual supervision is tested across natural images, scientific diagrams, documents, charts, tables, and multi-source reasoning tasks. A detailed visualization of the dataset distribution is provided in Appendix B. Representative visualizations of the constructed dataset are provided in Appendix C (Figures 5–20).

### 3.3 REINFORCEMENT LEARNING TRAINING

**Sequential Reasoning Formulation.** We formulate the reasoning process of DEFACTO as a Markov Decision Process (MDP), where the model interacts with the question and image in a sequential manner. At each step, the state $s_t$ encodes the multimodal context, including the input question, the image representation, and the history of previously predicted regions. The policy $\pi_\theta$ then outputs either a new bounding box that localizes question-relevant evidence or a special STOP token to terminate the process.

Formally, the state at step $t$ is defined as

$$s_t = \{q, f_v(I), B_{<t}\}, \tag{4}$$

where $q$ is the question, $f_v(I)$ the image representation, and $B_{<t}$ the set of bounding boxes predicted before step $t$. The rollout continues until STOP is emitted or the maximum step limit is reached, and the final answer is generated based on the accumulated trajectory.

**Reward Design.** To make training effective, we design three reward components. (1) *Answer Correctness Reward*: encourages correct answers in positive/random cases, rewards "Unknown" in counterfactual cases, and penalizes unsupported guesses. (2) *Format Consistency Reward*: ensures outputs strictly follow the required schema. (3) *Region Selection Coherence Reward*: promotes overlap with evidence regions $\mathcal{R}^+$ and penalizes overlap with irrelevant regions $\mathcal{R}^-$, with no reward in counterfactual cases.

The overall training signal combines these components into the composite reward in Eq. 5.

$$R = R_{\text{ans}} + \lambda_1 R_{\text{fmt}} + \lambda_2 R_{\text{sel}}, \tag{5}$$

**1. Answer Correctness Reward.** To enforce correct behavior across the three training forms, we define

$$R_{\text{ans}} = \begin{cases} \text{acc}(\hat{y}, y^*) - \underbrace{\gamma_{\text{unk}} \, \text{unk}(\hat{y})}_{\text{penalize "Unknown"}}, & t \in \{\text{pos}, \text{rand}\}, \\[2ex] \underbrace{\rho_{\text{unk}} \, \text{unk}(\hat{y})}_{\text{reward "Unknown"}} - \underbrace{\gamma_{\text{guess}} \left[1 - \text{unk}(\hat{y})\right]}_{\text{penalize guess}} - \underbrace{\gamma_{\text{corr}} \, \mathbf{1}[\hat{y} = y^*]}_{\text{penalize even if correct}}, & t = \text{cf}, \end{cases} \tag{6}$$

where $\text{acc}(\hat{y}, y^*) \in \{0, 1\}$ indicates answer correctness, and $\text{unk}(\hat{y}) \in \{0, 1\}$ indicates an "Unknown" response. Here $\gamma_{\text{unk}} > 0$ penalizes answering "Unknown" in positive or random cases, $\rho_{\text{unk}} > 0$ rewards "Unknown" in counterfactual cases, $\gamma_{\text{guess}} > 0$ penalizes any concrete guess in counterfactual cases, and $\gamma_{\text{corr}} > \gamma_{\text{guess}}$ applies an even stronger penalty when the model outputs the correct answer $y^*$ without access to evidence.

**2. Format Consistency Reward.** We encourage well-formed outputs and valid region indices selected from the prompt:

$$R_{\text{fmt}} = \begin{cases} \alpha, & \text{if output follows the required schema and indices are valid,} \\ 0, & \text{otherwise.} \end{cases} \tag{7}$$

Here, the "required schema" refers to the presence of `<think>`...`</think>` for the reasoning process, `<bbox>`...`</bbox>` for the predicted bounding boxes, and `<answer>`...`</answer>` for the final answer. In particular, the `<bbox>` field must contain well-formed bounding box coordinates in the format `[x1, y1, x2, y2]`, ensuring that the model explicitly grounds its predictions on localized visual regions.

**3. Region Selection Coherence Reward.** Let $B = \{\text{bbox}^1, \ldots, \text{bbox}^k\}$ be the set of bounding boxes predicted before STOP. We define the overlap scores with evidence regions $\mathcal{R}^+$ and irrelevant regions $\mathcal{R}^-$ as

$$\phi^+(b) = \max_{r \in \mathcal{R}^+} \text{IoU}(b, r), \quad \phi^-(b) = \max_{r \in \mathcal{R}^-} \text{IoU}(b, r).$$

The reward is then defined as

$$R_{\text{sel}} = \begin{cases} \beta_{\text{pos}} \frac{1}{|B|} \sum_{b \in B} \phi^+(b) \; - \; \beta_{\text{neg}} \frac{1}{|B|} \sum_{b \in B} \phi^-(b), & t \in \{\text{pos}, \text{rand}\}, \, B \neq \varnothing, \\ - \gamma_\varnothing, & t \in \{\text{pos}, \text{rand}\}, \, B = \varnothing, \\ 0, & t = \text{cf}. \end{cases} \quad (8)$$

with $\beta_{\text{pos}}, \beta_{\text{neg}}, \gamma_\varnothing > 0$.

The full training dynamics of each reward component, along with the corresponding hyperparameter settings, are provided in the Appendix (see Table 9).

**Training Strategy.** Unlike prior works that require a supervised warm-up stage, we directly fine-tune Qwen2.5-VL with reinforcement learning, using Group Relative Policy Optimization (GRPO) and the composite reward in Eq. 6 and Eq. 8. GRPO compares multiple rollouts within a group and rewards each according to its improvement over the group average, eliminating the need for a value network and reducing variance. The objective is defined as:

$$\mathcal{L}_\pi(\theta) \; = \; \mathbb{E}_i \left[ \frac{\pi_\theta(\tau^{(i)})}{\pi_{\theta_{\text{old}}}(\tau^{(i)})} \left( R(\tau^{(i)}) - \frac{1}{M} \sum_{j=1}^{M} R(\tau^{(j)}) \right) \right], \quad (9)$$

where we set the group size $M = 4$ to balance stability and exploration during training.

## 4 EXPERIMENT

### 4.1 SETUP

**Baselines.** We compare DEFACTO against a broad set of recent approaches that explicitly incorporate visual reasoning into multimodal language models. Specifically, we include the QWEN2.5-VL Bai et al. (2025), a strong pretrained backbone widely used for visual understanding; VI-CROP Zhang et al. (2025a), which improves small-object perception via inference-time cropping; GRIT Fan et al. (2025), which integrates grounded reasoning traces through reinforcement learning; DEEPEYES Zheng et al. (2025), which incentivizes models to call visual tools during reasoning; and VISUAL-SR1 Li et al. (2025b), which enhances step-by-step visual reasoning with self-refinement. This selection covers both state-of-the-art backbones and recent "thinking with images" algorithms for visual reasoning.

**Benchmarks.** Our evaluation spans a diverse collection of visual reasoning benchmarks. For general-purpose VQA, we use OKVQA Marino et al. (2019), VQAv2 Goyal et al. (2017), GQA Hudson & Manning (2019), VizWiz Gurari et al. (2018), ScienceQA Lu et al. (2022), and VSR Liu et al. (2023). For document- and structure-centric evaluation, we adopt DocVQA Mathew et al. (2021), ChartQA Masry et al. (2022), InfoVQA Mathew et al. (2022), DeepForm Svetlichnaya (2020), Kleister KLC Stanisławek et al. (2021), and WikiTableQuestions (WTQ) Pasupat & Liang (2015). To test text-intensive reasoning, we include TextVQA Singh et al. (2019), AI2D Kembhavi et al. (2016), and STVQA Biten et al. (2019). We further evaluate on additional benchmarks, including OCRBench Liu et al. (2024c), MMstar Chen et al. (2024a), MMMU Yue et al. (2024), MMB$_{1.1}$ Liu et al. (2024b), and POPE Li et al. (2023b), with detailed results reported in Appendix E (Table 6). In addition, to more rigorously assess reasoning faithfulness, we constructed a manually annotated test set of 1,000 images, where annotators labeled the regions most relevant to each

Table 1: Results on General VQA Benchmarks (accuracy, %). $\Delta$ indicates improvements of DeFacto over Qwen2.5-VL 7B.

| Model | Backbone | VQAv2 | OKVQA | GQA | SciQA | VizWiz | VSR |
|---|---|---|---|---|---|---|---|
| Qwen2.5-VL | Qwen2.5-VL-7B | 57.3 | 54.5 | 41.3 | 85.3 | 37.7 | 2.2 |
| ViCrop | LLaVA-1.5 (Vicuna-7B) | 76.5 | 60.7 | 61.0 | 88.2 | 64.4 | 65.4 |
| GRIT | Qwen2.5-VL-3B | 71.5 | 55.4 | 59.3 | 60.5 | 46.3 | 61.0 |
| DeepEyes | Qwen2.5-VL-7B | – | 46.9 | 47.3 | 59.2 | 25.1 | 27.3 |
| Visual-SR1 | Qwen2.5-VL-7B | 71.5 | 45.1 | 58.5 | **88.6** | 32.0 | 62.3 |
| DeFacto (ours) | Qwen2.5-VL-7B | **79.7** | **68.0** | **70.1** | 88.2 | 64.5 | **70.3** |
| $\Delta$ (vs Qwen2.5-VL 7B) | – | – | +22.4 | +13.5 | +28.8 | +2.9 | +26.8 | +68.1 |

Table 2: Results on Document VQA and Scene Text-centric Benchmarks (accuracy, %). $\Delta$ indicates improvements of DeFacto over Qwen2.5-VL 7B.

| Model | Document VQA | | | | | | Scene Text-centric | | |
|---|---|---|---|---|---|---|---|---|---|
| | DocVQA | ChartQA | InfoVQA | DeepForm | KLC | WTQ | STVQA | TextVQA | AI2D |
| Qwen2.5-VL | 84.4 | 77.8 | 66.0 | 30.3 | 35.9 | 63.9 | 64.9 | 71.0 | 71.2 |
| GRIT | 76.4 | 68.7 | 49.1 | 15.8 | 19.9 | 35.7 | 71.3 | **73.4** | 77.2 |
| ViCrop | 33.7 | 52.5 | 54.9 | 21.9 | 33.4 | 54.3 | 74.0 | 63.4 | 69.2 |
| DeepEyes | 66.8 | 44.4 | 42.3 | – | 33.1 | 54.6 | 48.9 | 39.9 | 38.5 |
| Visual-SR1 | 82.3 | 73.8 | 75.1 | **52.4** | **39.6** | 72.2 | 60.2 | 69.2 | 71.5 |
| DeFacto (ours) | **85.8** | **82.4** | **76.9** | 51.8 | 37.6 | **74.5** | **74.1** | **73.4** | **79.0** |
| $\Delta$ (vs Qwen2.5-VL 7B) | +1.4 | +4.6 | +10.9 | +21.5 | +1.7 | +10.6 | +9.2 | +2.4 | +7.8 |

question. The set contains 60% general VQA samples and 40% text-centric VQA samples. This auxiliary dataset allows us to directly measure whether models ground their reasoning on the correct evidence rather than relying on spurious priors.

**Training Configuration.** We train all models with the AdamW optimizer using a learning rate of $1 \times 10^{-6}$, $(\beta_1, \beta_2) = (0.9, 0.999)$, and $\epsilon = 1 \times 10^{-8}$. Training is performed with a global batch size of 8 and micro-batch size of 1 per GPU, combined with gradient accumulation steps of 2. Gradients are clipped to a maximum norm of 1.0 to ensure stability. We enable BF16 precision training. All experiments are conducted on 8 NVIDIA H100 GPUs with 80GB memory each, and models are trained for one epoch over the collected dataset.

## 4.2 MAIN RESULTS

**Results on General VQA Benchmarks.** Table 1 compares DEFACTO with recent visual reasoning and thinking with images baselines on six widely used benchmarks. DEFACTO achieves state-of-the-art performance across the board, outperforming the strongest competing method, Vi-Crop, by clear margins. In particular, it improves over ViCrop by +3.2% on VQAv2, +7.3% on OKVQA, and +9.1% on GQA, demonstrating stronger compositional and commonsense reasoning. On perception-heavy datasets, DEFACTO also shows advantages: it slightly surpasses Visual-SR1 on SciQA while matching ViCrop on VizWiz, and it exceeds all baselines on VSR by +4.9%, confirming its robustness under visually complex or noisy conditions. These consistent gains over the best-performing alternatives highlight the effectiveness of counterfactual training in enforcing evidence-grounded reasoning.

**Performance on Document and Text-centric Benchmarks.** As shown in Table 2, DEFACTO also leads on document-style and scene text-centric benchmarks. It surpasses the strongest alternatives by notable margins, including +1.4% over Qwen2.5-VL on DocVQA, +4.6% on ChartQA, and +1.8% over Visual-SR1 on InfoVQA. On DeepForm, although Visual-SR1 achieves the best score, DEFACTO remains highly competitive with a close result of 51.8%. Similarly, while GRIT ties for the highest score on TextVQA, DEFACTO delivers the best overall performance across all text-centric tasks, including a +13.9% gain over Visual-SR1 on STVQA and a +1.8% improvement over GRIT on AI2D. These results confirm that DEFACTO not only consistently outperforms the strongest existing methods but also maintains competitive accuracy in the few cases where another

baseline achieves the top result, establishing a new state of the art in document and OCR-centric reasoning tasks.

Table 3: Comparison of reasoning faithfulness across models on the 1k faithfulness validation set.

| Model | mAP | AP50 | AP75 | Accuracy (%) |
|---|---|---|---|---|
| GRIT | 0.0 | 0.0 | 0.0 | 73.7 |
| DeepEyes | 0.0 | 2.2 | 0.9 | 44.0 |
| Qwen2.5-VL + GRPO | 20.4 | 28.8 | 18.9 | 65.0 |
| DeFacto (ours) | **30.6** | **36.1** | **24.8** | **79.4** |

**Faithful reasoning evaluation.** Table 3 complements the qualitative examples in Fig. 1 with a large-scale quantitative evaluation on our 1k-image human-annotated validation set. The results reveal several consistent patterns. GRIT achieves relatively high answer accuracy, but its grounding metrics remain nearly zero, suggesting that it has learned to rely on shortcuts rather than visually grounded reasoning. Such shortcut behaviors allow the model to answer correctly without identifying the relevant evidence, which is undesirable for faithful multimodal reasoning. DeepEyes shows a different failure pattern: while it learns to invoke visual tools more actively, it does not reliably learn where to focus. As a result, its predicted regions often diverge from the true evidence, leading to low IoU-based AP despite occasional correct answers. GRPO improves over these baselines by producing more stable predictions, but its evidence regions are still coarse and only partially aligned with the annotated visual cues. In contrast, DeFacto achieves the highest grounding fidelity across all metrics. The counterfactual reward and region-level constraints guide the model toward consistently selecting the evidence required for its predicted answer, resulting in reasoning trajectories that better reflect the underlying visual content.

## 4.3 ABLATION STUDY

We compare four training settings on Qwen2.5-VL 7B. (i) **SFT (no CF)**: trained only on original data. (ii) **SFT (CF alignment)**: trained on original + counterfactual data, but counterfactuals are supervised only with the "Unknown" label, together with random-masking. (iii) **GRPO (no CF reward)**: "No CF reward" means that GRPO training that uses only the first term of Eq. 6 (answer correctness) and the format reward, without the counterfactual answer constraint or the region-selection coherence term. (iv) **DeFacto (full)**: our complete framework with all three rewards.

Table 4: Ablation results on representative benchmarks (accuracy, %). $\Delta$ indicates improvements of DeFacto over baselines.

| Model Variant | VQAv2 | OKVQA | SciQA | VSR | DocVQA | TextVQA |
|---|---|---|---|---|---|---|
| Qwen2.5-VL (SFT, no CF) | 61.2 | 42.0 | 82.7 | 54.5 | 51.9 | 56.0 |
| Qwen2.5-VL (SFT, CF alignment) | 66.5 | 55.7 | 84.7 | 53.7 | 84.3 | 73.0 |
| Qwen2.5-VL (GRPO, no CF reward) | 70.4 | 56.9 | 85.9 | 58.4 | 85.4 | 72.8 |
| DeFacto (CF reward + GRPO) | **79.7** | **68.0** | **88.2** | **70.3** | **85.8** | **73.4** |
| $\Delta$ (vs GRPO no CF reward) | +9.3 | +11.1 | +2.3 | +11.9 | +0.4 | +0.6 |

**Effect of Counterfactual Supervision.** The first two rows show that introducing counterfactual data with abstention alignment improves over standard SFT, with clear gains such as +5.3% on VQAv2 and +13.7% on OKVQA. This suggests that counterfactual supervision effectively reduces spurious correlations and strengthens evidence alignment.

**Effect of Reinforcement Learning.** GRPO without counterfactual rewards further boosts reasoning, e.g., +4.7% on VSR over SFT no CF. DeFacto achieves the best results overall, with additional gains of +9.3% on VQAv2, +11.1% on OKVQA, and +11.9% on VSR compared to GRPO. Even on DocVQA and TextVQA, improvements (+0.4%, +0.6%) remain consistent, confirming the importance of counterfactual rewards for robust, region-faithful reasoning.

**Faithfulness Evaluation.** In the experiment, we compared different training variants under the same setup, using the same data, backbone, and hyperparameters. The results, shown in Table 5, indicate that DeFacto outperforms all other variants in terms of mAP, AP50, AP75, and accuracy, demonstrating that our approach effectively combines correct evidence selection and accurate answering.

Table 5: Performance of Different Training Variants on the 1k-annotated Validation Set

| Model Variant | mAP | AP50 | AP75 | Accuracy (%) |
|---|---|---|---|---|
| Qwen2.5-VL (Base) | 2.3 | 1.2 | 1.5 | 55.1 |
| Qwen2.5-VL + SFT (no CF) | 15.7 | 13.9 | 12.8 | 61.4 |
| Qwen2.5-VL + SFT (CF alignment) | 18.9 | 16.4 | 15.7 | 63.8 |
| Qwen2.5-VL + GRPO (no CF reward) | 20.4 | 28.8 | 18.9 | 65.0 |
| **DeFacto (ours)** | **30.6** | **36.1** | **24.8** | **79.4** |

## 5 VISUALIZATION AND ERROR ANALYSIS

We first visualize and analyze several failure cases to better understand the limitations of DeFacto. Our errors mainly fall into four categories: (1) semantic ambiguity in spatial expressions (Figure 3a), such as interpreting "on top of" too literally; (2) unclear or subjective attribute definitions (Figure 3b), such as the notion of "large"; (3) ambiguous comparative reasoning (Figure 3c), where concepts like "higher than" admit multiple valid interpretations; and (4) confusion in fine-grained human action recognition (Figure 3d).

a) What is located on top of the shoe?
Ours (Defacto): cat
GT: shoe lace

b) Does the black fireplace look large?
Ours (Defacto): yes
GT: no

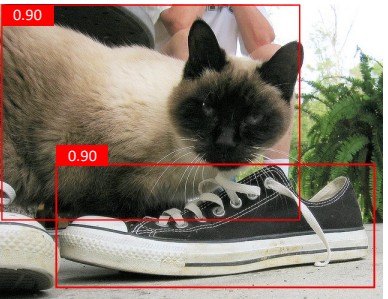

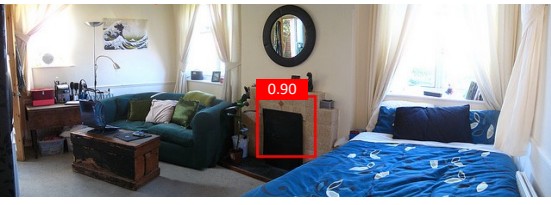

d) What is the person to the left of the catcher doing?
Ours (Defacto): standing
GT: looking up

c) What is higher than the window?
Ours (Defacto): pipe
GT: ceiling

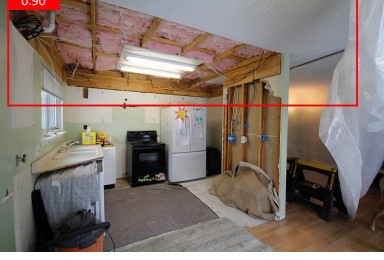

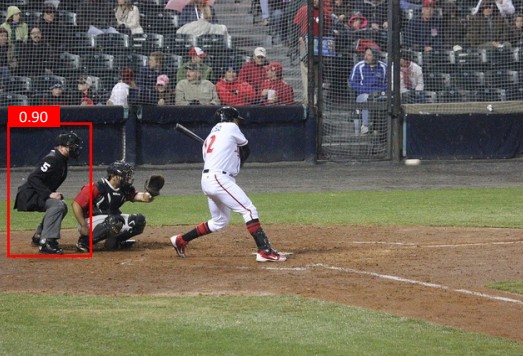

Figure 3: Four representative failure cases illustrating different types of semantic and perceptual ambiguities.

Beyond failure cases, we further visualize comparisons between DeFacto and standard GRPO to assess the quality of their reasoning paths. As shown in Appendix D (Figures 21–28), DeFacto consistently selects more accurate and semantically aligned evidence regions.

## ETHICS STATEMENT

This work does not involve human subjects, sensitive personal data, or applications with foreseeable ethical risks. All experiments are conducted on publicly available benchmarks or our automatically constructed counterfactual dataset, which contains no personally identifiable information. We therefore believe this research poses no ethical concerns.

## REPRODUCIBILITY STATEMENT

We have made every effort to ensure reproducibility of our results. The full training pipeline, including dataset construction, reward design, and reinforcement learning setup, is described in detail in the main paper and appendix. Our implementation is based on the open-source open_r1 OpenAI (2025) repository, which we modified to incorporate our counterfactual dataset, reward functions, and GRPO training. The source code is provided in the supplementary materials to facilitate reproduction. Due to anonymity requirements and the large size of the dataset, we are unable to release the full dataset at this stage; however, it will be made publicly available upon acceptance. In the meantime, partial dataset visualizations are included in Appendix H to illustrate the construction process and provide qualitative insights.

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

APPENDIX

## A  ADDITIONAL RESULTS

Table 6 provides an extended comparison of *DeFacto* with both closed-source and publicly available vision-language models on additional benchmarks, including **OCRBench**, **MMstar**, **MMMU**, **MMB$_{1.1}$**, and **POPE**. As shown, DeFacto achieves the best score on OCRBench (871) and competitive performance across the remaining benchmarks. In particular, our method surpasses most open-source models by a clear margin and remains close to strong closed-source systems such as GPT-4o. These results further demonstrate that DeFacto effectively balances answer accuracy and reasoning faithfulness across diverse evaluation settings.

In addition, we also evaluate DeFacto on the MSTI 2.0 dataset Chen et al. (2024d), which jointly assesses detection, entity recognition, and answer generation. As shown in Table 7, DeFacto consistently improves over both the zero-shot Qwen2.5-VL model and the GRPO-CoT variant across EM, F1, and AP metrics. This indicates that DeFacto's counterfactual supervision also benefits structured multimodal tasks requiring fine-grained grounding.

Table 6: Comparison with SoTA models on Various Benchmarks.

| Model | OCRB | MMstar | MMMU | MMB$_{1.1}$ | POPE |
|---|---|---|---|---|---|
| **Closed-Source Models** | | | | | |
| GPT-4o-0513  Microsoft (2024) | 736 | **63.9** | **69.2** | **82.2** | - |
| GPT-4V  Team | 656 | 56.0 | 61.7 | 79.8 | - |
| Gemini-1.5-Pro Team et al. (2024) | 754 | - | 62.2 | - | - |
| **Publicly Available Models** | | | | | |
| LLaVa-OneVision-0.5B Li et al. (2024) | 565 | 37.7 | 31.4 | 50.3 | - |
| InternVL2-1B Team (2024) | 754 | 45.7 | 36.7 | 59.7 | - |
| Eagle2-1B Li et al. (2025a) | 767 | 48.5 | 38.8 | 63.0 | - |
| InternVL2-2B Team (2024) | 784 | 50.1 | 36.3 | 69.6 | - |
| Eagle2-2B Li et al. (2025a) | 818 | 56.4 | 43.1 | 74.9 | - |
| InternVL2-8B Team (2024) | 794 | 60.9 | 51.8 | 79.4 | - |
| MiniCPM-V2.6 Hu et al. (2024) | 852 | 57.5 | 49.8 | 78.0 | - |
| LLaVA-One-Vision-7B  Li et al. (2024) | 622 | 61.7 | 48.8 | 80.9 | - |
| InternVL2-26B Team (2024) | 825 | 61.0 | 50.7 | 81.2 | - |
| LLaMa-3.2-90B-Vision  Grattafiori et al. (2024) | 783 | 55.3 | 60.3 | 77.3 | - |
| HyViLMZhu et al. (2024) | 596 | - | 41.8 | 76.6 | - |
| Eagle2-9B Li et al. (2025a) | 868 | 62.6 | 56.1 | 80.6 | - |
| **Thinking with Images** | | | | | |
| GRIT  Fan et al. (2025) | 322 | 36.3 | 17.1 | 9.7 | 85.7 |
| ViCrop Zhang et al. (2025a) | 233 | 33.1 | 26.1 | 51.7 | 87.3 |
| DeepEyes  Zheng et al. (2025) | 636 | 43.6 | 44.1 | 29.4 | 87.7 |
| Visual-SR1  Li et al. (2025b) | 449 | 62.8 | 57.2 | 77.4 | 86.0 |
| Chain-of-focus Zhang et al. (2025b) | 632 | 58.1 | 46.1 | 75.3 | 88.4 |
| Pixel Reasoner Su et al. (2025) | 597 | 62.9 | 52.5 | 78.5 | 87.8 |
| **DeFacto-7B (ours)** | **871** | 63.2 | 56.6 | 81.2 | **88.6** |

Table 7: Comparison of Qwen2.5-VL variants and DeFacto on the MSTI2.0 dataset.

| | Dev | | | Test | | |
|---|---|---|---|---|---|---|
| | EM | F1 | AP | EM | F1 | AP |
| Qwen2.5-VL (Zero-shot) | 22.2 | 1.1 | 1.2 | 18.6 | 2.6 | 0.7 |
| Qwen2.5-VL (GRPO-CoT) | 24.3 | 1.4 | 3.3 | 19.7 | 3.1 | 1.2 |
| **DeFacto (ours)** | **28.4** | **1.6** | **4.1** | **23.1** | **4.5** | **2.5** |

## B    DATASET DISTRIBUTION VISUALIZATION

The full distribution of the 100k training samples is illustrated in Figure 4. The dataset is divided into three major groups: general VQA (47.67%), scene text-centric VQA (22.05%), and document-oriented VQA (30.28%).

General VQA primarily corresponds to natural-image domains such as everyday scenes and objects, providing perception-heavy signals. Scene text-centric VQA consists largely of OCR-focused questions in real-world contexts, capturing text in cluttered environments. Document-oriented VQA covers structured layouts including documents, charts, tables, and forms, emphasizing fine-grained text extraction and layout reasoning. This mixture ensures broad coverage across both natural-image and document-like domains. By preventing dominance from any single modality and exposing the model to heterogeneous visual structures, the dataset encourages stronger domain generalization and reduces reliance on narrow visual priors. Such diversity is particularly important for improving robustness in downstream tasks that span multiple visual domains.

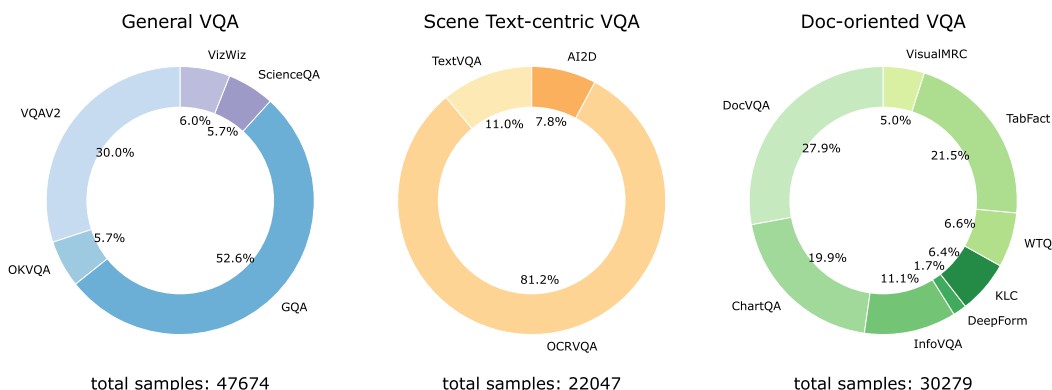

Figure 4: VQA Data Distribution across different categories: General VQA, Scene-text VQA, and Document-oriented VQA.

## C    VISUALIZATION OF COUNTERFACTUAL DATASET

In this section, we provide visualizations of the constructed counterfactual dataset. Each sample consists of three views:

**(a) Original**: the original unmodified image.

**(b) Original_smask**: the image with task-relevant (key) regions masked out.

**(c) Original_rmask**: the image with task-irrelevant regions masked out.

Figures 5–20 show representative examples from the dataset.

Question: What type of boat is this? Answer:

Answer: barge

**(a) Original**      **(b) Original_smask**      **(c) Original_rmask**

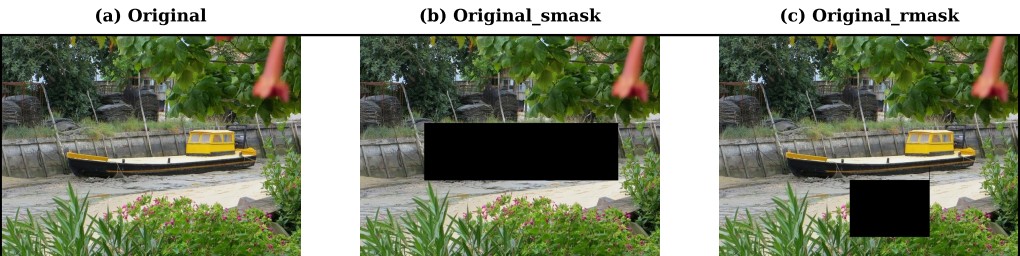

Figure 5: Visualization Example 1

Question: What is the red line represents? Answer:

Answer: Share of women who prefer a male boss

**(a) Original**      **(b) Original_smask**      **(c) Original_rmask**

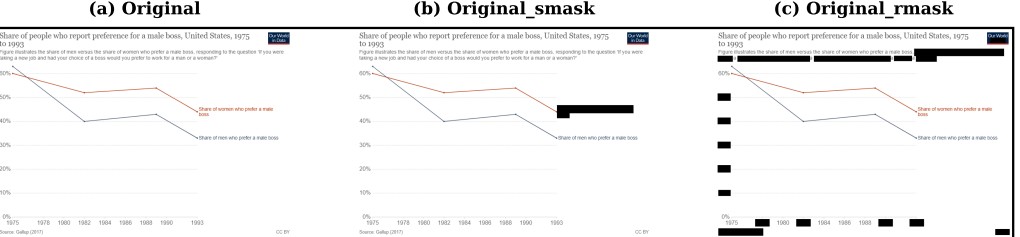

Figure 6: Visualization Example 2

Question: Which kind of food is to the left of the fork? Answer:

Answer: salad

**(a) Original**      **(b) Original_smask**      **(c) Original_rmask**

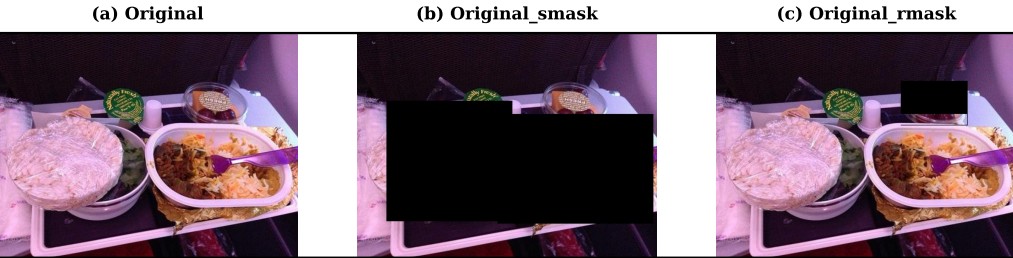

Figure 7: Visualization Example 3

Question: What is the water in front of? Answer:

Answer: trees

**(a) Original**    **(b) Original_smask**    **(c) Original_rmask**

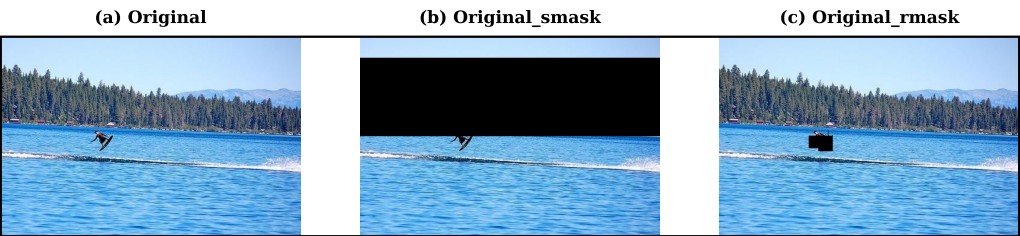

Figure 8: Visualization Example 4

Question: What is the frame made of? Answer:

Answer: wood

**(a) Original**    **(b) Original_smask**    **(c) Original_rmask**

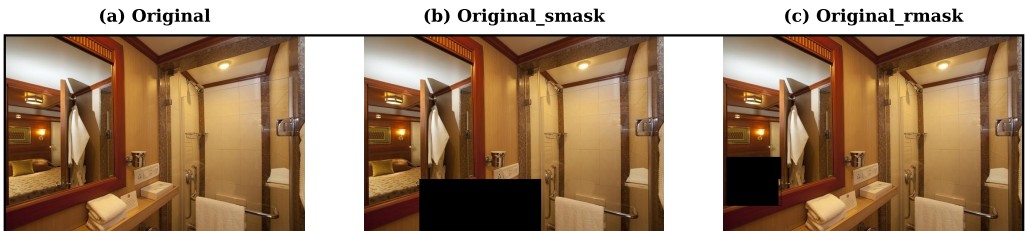

Figure 9: Visualization Example 5

Question: Is the truck parked straight on a driveway? Answer:

Answer: no

**(a) Original**    **(b) Original_smask**    **(c) Original_rmask**

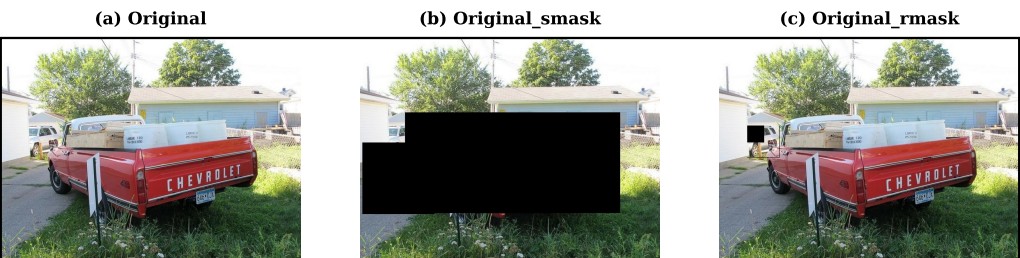

Figure 10: Visualization Example 6

Question: Which kind of animal is standing on the hay? Answer:

Answer: cow

**(a) Original**            **(b) Original_smask**            **(c) Original_rmask**

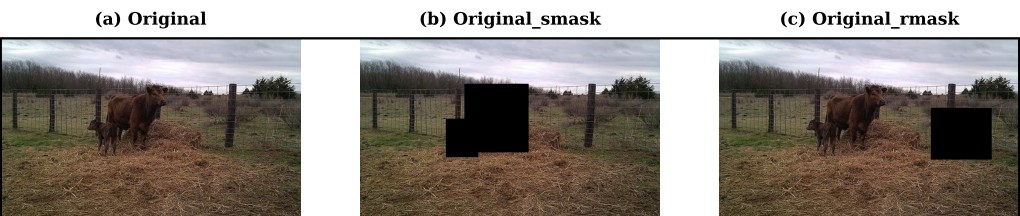

Figure 11: Visualization Example 7

Question: What is the total direct enterprise investment of Ireland (in euros) in start-ups in 2015? Answer:

Answer: 31m

**(a) Original**            **(b) Original_smask**            **(c) Original_rmask**

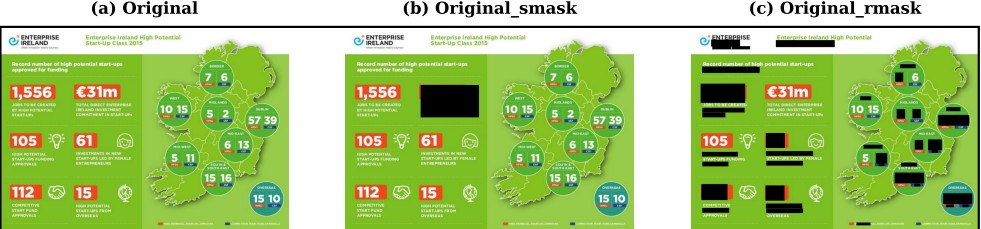

Figure 12: Visualization Example 8

Question: what is the name of the wine? Answer:

Answer: italica

**(a) Original**            **(b) Original_smask**            **(c) Original_rmask**

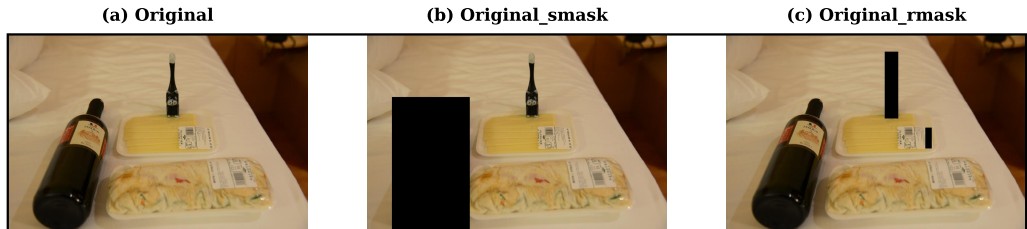

Figure 13: Visualization Example 9

Question: Who is wearing a vest? Answer:

Answer: woman

(a) Original          (b) Original_smask          (c) Original_rmask

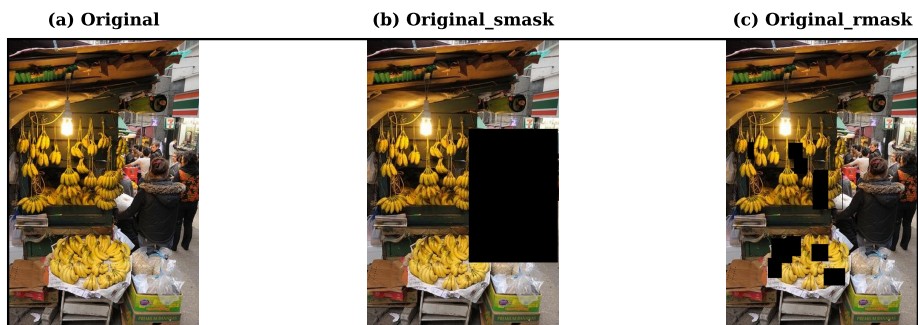

Figure 14: Visualization Example 10

Question: what number gate is this? Answer:

Answer: 97

(a) Original          (b) Original_smask          (c) Original_rmask

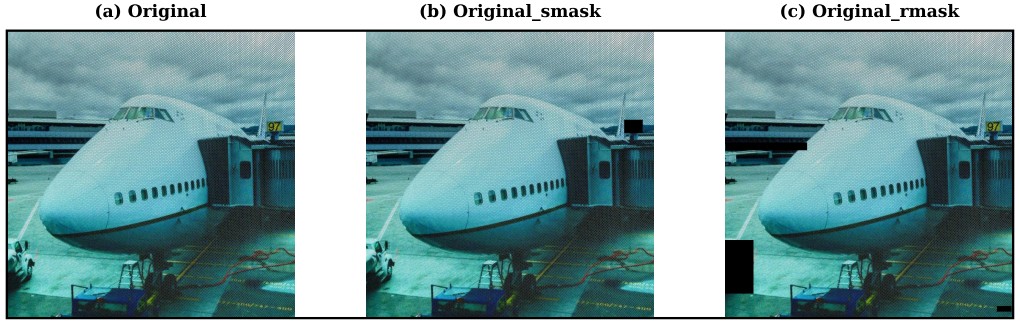

Figure 15: Visualization Example 11

Question: where is this bus going? Answer:

Answer: 8mt st vincent

(a) Original          (b) Original_smask          (c) Original_rmask

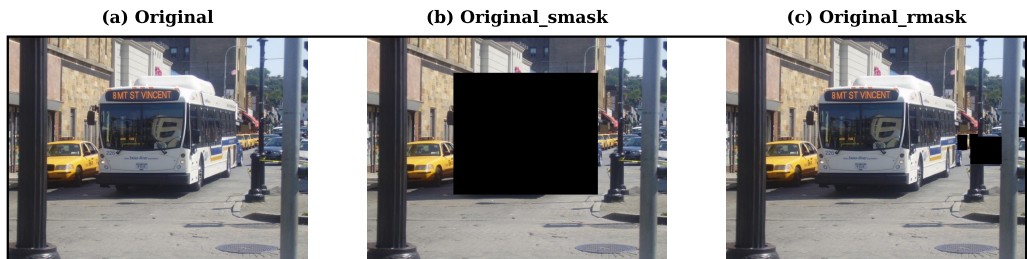

Figure 16: Visualization Example 12

Question: What color is the bag? Answer:

Answer: green

(a) Original  (b) Original_smask  (c) Original_rmask

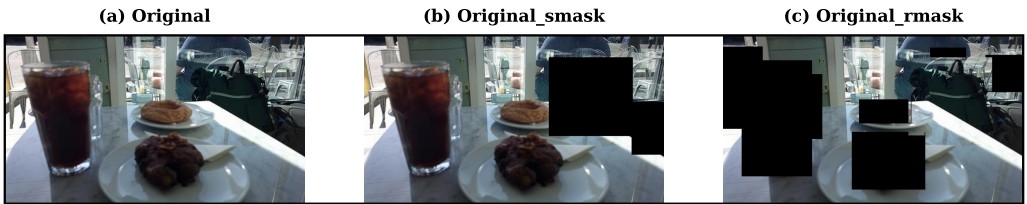

Figure 17: Visualization Example 13

Question: What was the GDP per capita in Madagascar in 2020? Answer:

Answer: 501.76

(a) Original  (b) Original_smask  (c) Original_rmask

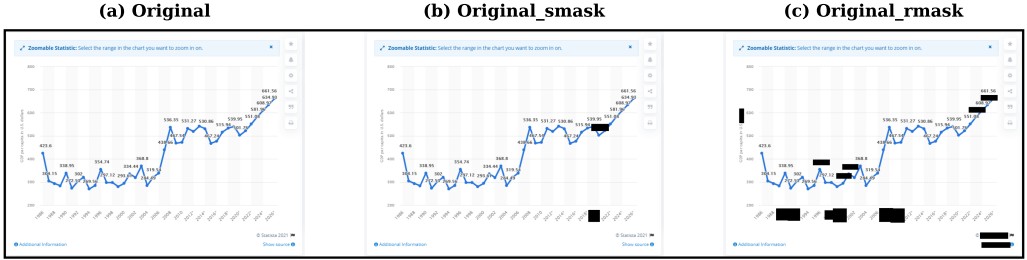

Figure 18: Visualization Example 14

Question: What company had a share of 16.5 percent of the world liner fleet? Answer:

Answer: Mediterranean Shg Co

(a) Original  (b) Original_smask  (c) Original_rmask

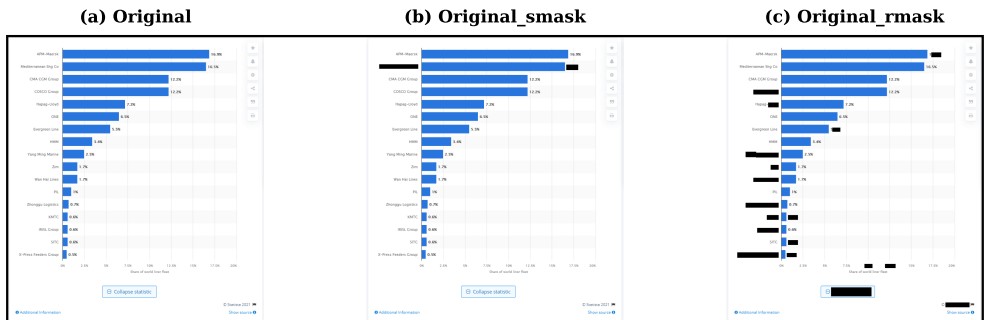

Figure 19: Visualization Example 15

Question: What hairstyle does the woman have? Answer:

Answer: ponytail

**(a) Original**     **(b) Original_smask**     **(c) Original_rmask**

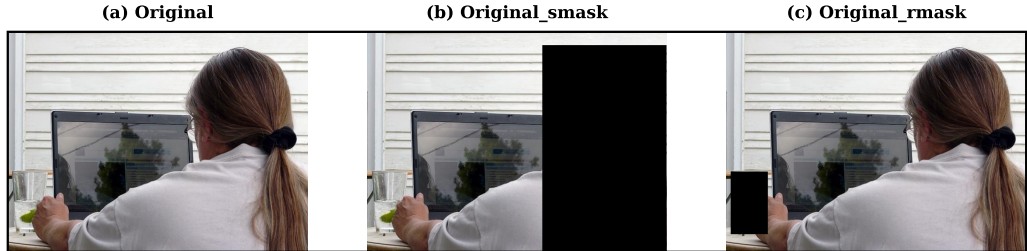

Figure 20: Visualization Example 16

## D    VISUALIZATION EXAMPLES

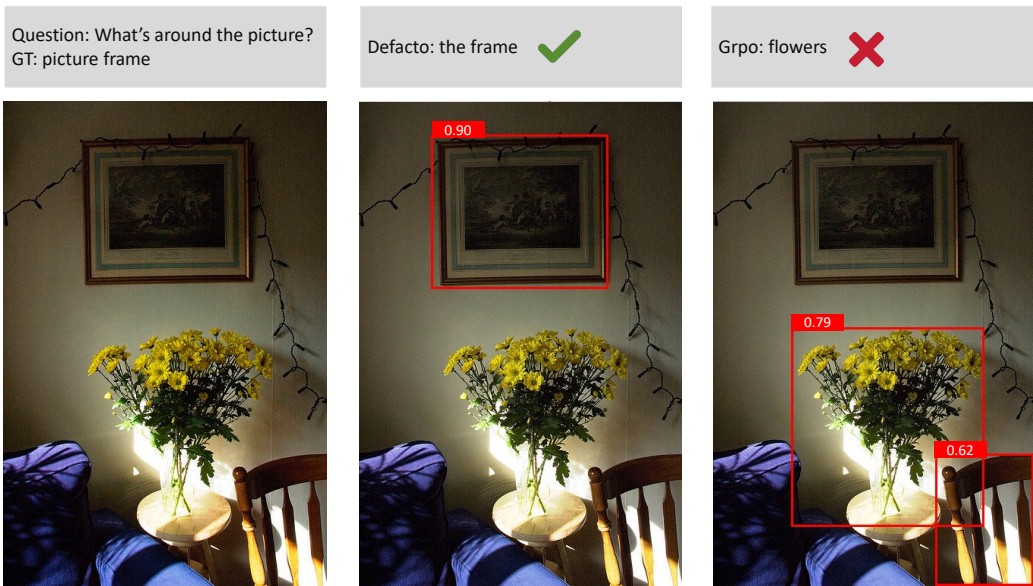

Question: What's around the picture?
GT: picture frame

Defacto: the frame ✔

Grpo: flowers ✘

Figure 21: Visualization examples comparing DeFACTO and standard GRPO (Example 1)

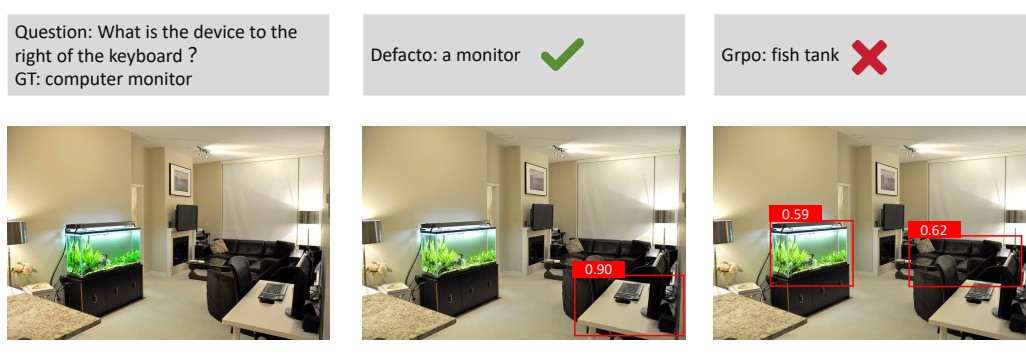

Question: What is the device to the right of the keyboard ?
GT: computer monitor

Defacto: a monitor ✔

Grpo: fish tank ✘

Figure 22: Visualization examples comparing DeFACTO and standard GRPO (Example 2)

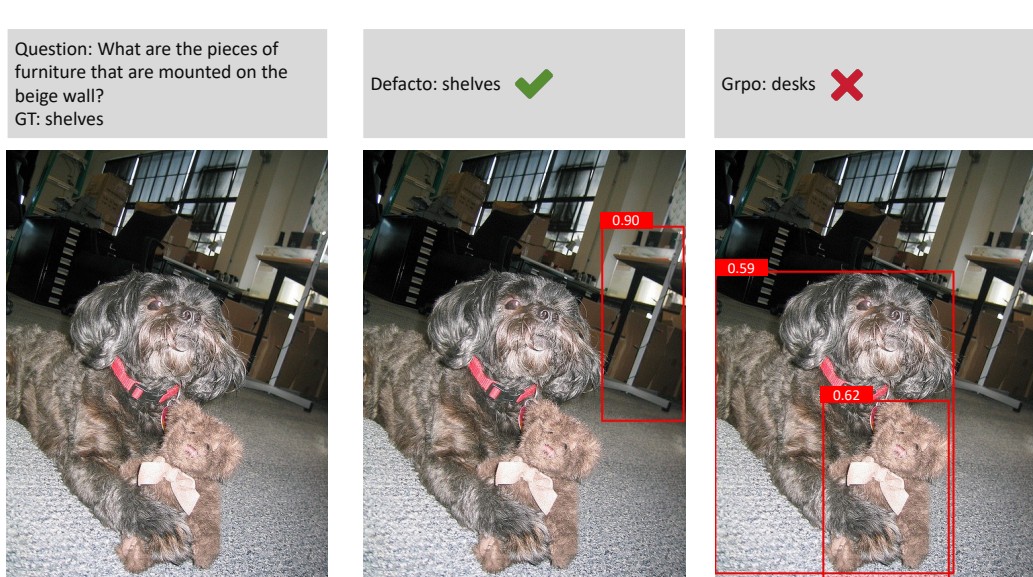

Figure 23: Visualization examples comparing DeFACTO and standard GRPO (Example 3)

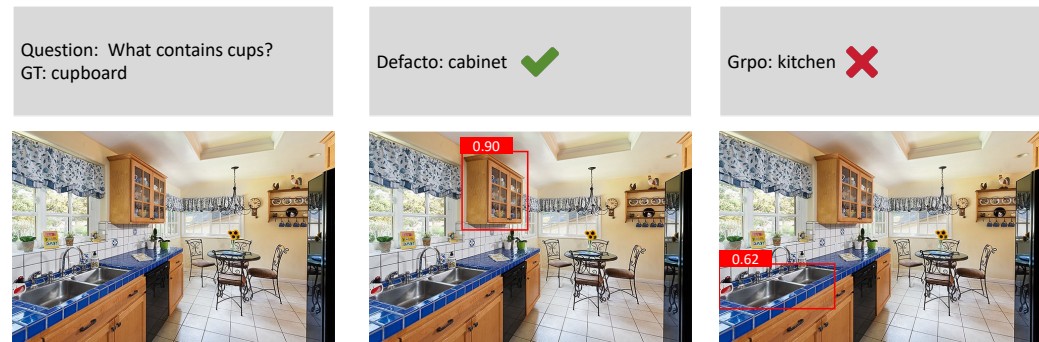

Figure 24: Visualization examples comparing DeFACTO and standard GRPO (Example 4)

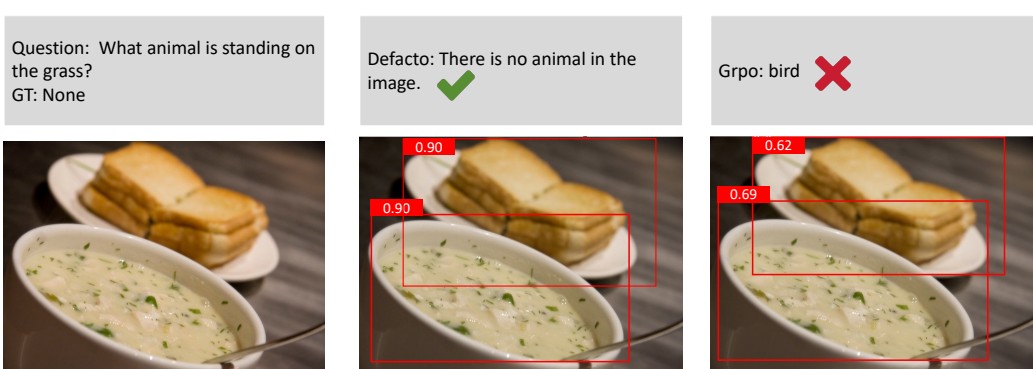

Figure 25: Visualization examples comparing DeFACTO and standard GRPO (Example 5)

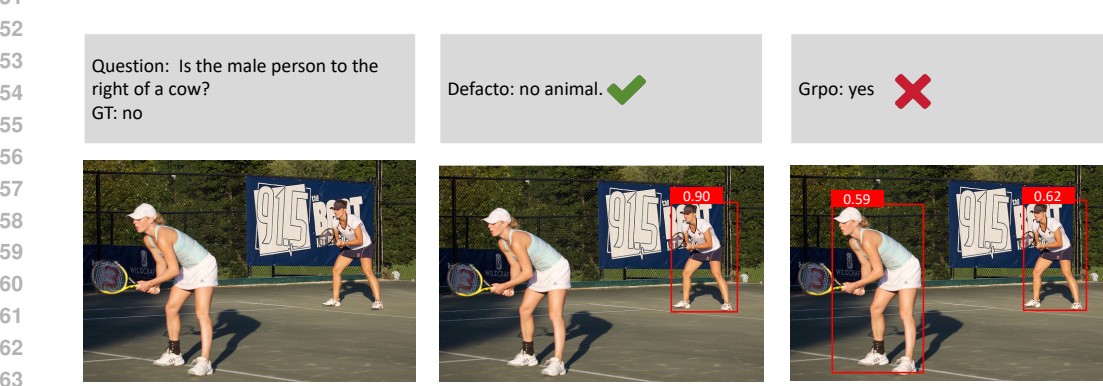

Figure 26: Visualization examples comparing DeFACTO and standard GRPO (Example 6)

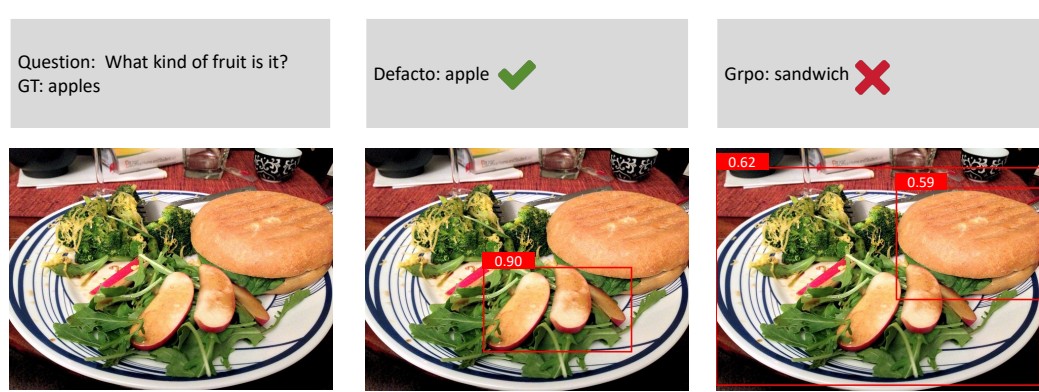

Figure 27: Visualization examples comparing DeFACTO and standard GRPO (Example 7)

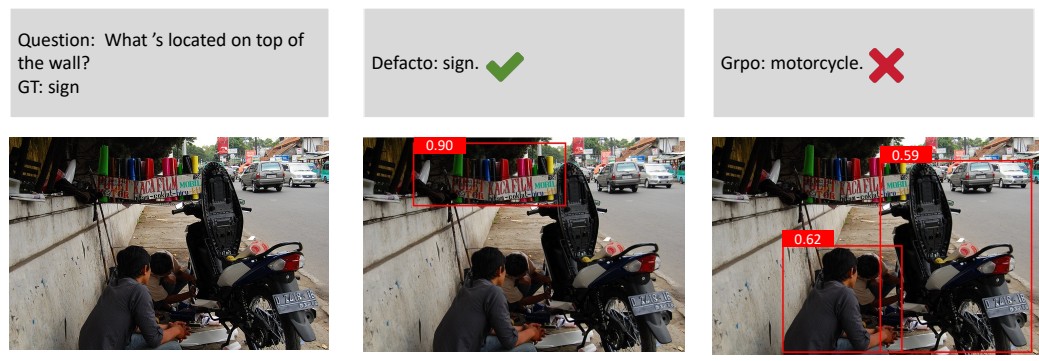

Figure 28: Visualization examples comparing DeFACTO and standard GRPO (Example 8)

# E SYSTEM PROMPT EXAMPLE

---

**System Prompt**

```
Please answer my question based on the image I have provided. Identify the region in the
image that is most relevant to the question and provide bounding box coordinates.
Output requirements:
1. The bounding boxes must be wrapped in <bbox> ... </bbox> tags.
2. The thinking process must be wrapped in <think> ... </think> tags.
3. The final answer must be wrapped in <answer> ... </answer> tags.
Format of the response (must strictly follow this structure): <bbox>['Position': [x1, y1,
x2, y2], 'Confidence': number] </bbox> <think> ... </think> <answer>...</answer>
The current question is:
```

---

# F TRAINING CONFIGURATION DETAILS

Table 8 summarizes the key hyperparameters used in our experiments.

Table 8: Training configuration.

| Parameter | Value |
|---|---|
| Optimizer | AdamW |
| Learning rate | $1 \times 10^{-6}$ |
| Adam betas | (0.9, 0.999) |
| Adam $\epsilon$ | $1 \times 10^{-8}$ |
| Weight decay | 0.0 |
| Precision | BF16 (FP16 disabled) |
| Batch size (global) | 8 |
| Micro batch size / GPU | 1 |
| Gradient accumulation steps | 2 |
| Gradient clipping | 1.0 |
| ZeRO optimization | Stage 3 with CPU offloading |
| Overlap communication | Enabled |
| Pinned memory | Enabled |
| Steps per print | inf |
| Wall clock breakdown | False |
| Hardware | $8 \times$ NVIDIA H100 (80GB) |
| Epochs | 1 |

# G PROMPT FOR EVALUATION

During evaluation, we employed `Qwen3` as the judge model to score the generated answers. The following prompt was used to guide the evaluation process:

---

**Evaluation Prompt**

```
"prompt": "Human: You are responsible for proofreading the answers, you need to give
a score to the model's answer by referring to the standard answer, based on the given
question. The full score is 1 point and the minimum score is 0 points. Please output the
score in the format <score>: The evaluation criteria require that the closer the model's
answer is to the standard answer, the higher the score. Note that the standard answer may
be a list containing multiple possible correct answers."
```

---

# H REWARD HYPERPARAMETERS

Table 9 reports the hyperparameter settings used in our reward design (Eq. 5). These values were tuned to balance the contributions of the three reward components: answer correctness ($R_{ans}$), format consistency ($R_{fmt}$), and region selection coherence ($R_{sel}$). The settings reflect the following intuition: $\gamma_{corr}$ is set larger than $\gamma_{guess}$ to penalize counterfactual "lucky guesses" more severely, $\rho_{unk}$ rewards

correct abstentions in counterfactual cases, and $\beta_{\mathrm{pos}}$ is emphasized to encourage stronger alignment with evidence regions in positive/random cases. Together with the weighting $(\lambda_1, \lambda_2)$.

Table 9: Reward hyperparameter settings for the composite reward in Eq. 5.

| Component | Parameter | Value |
|---|---|---|
| Answer Correctness ($R_{\mathrm{ans}}$) | $\gamma_{\mathrm{unk}}$ | 0.2 |
| | $\rho_{\mathrm{unk}}$ | 1.0 |
| | $\gamma_{\mathrm{guess}}$ | 0.8 |
| | $\gamma_{\mathrm{corr}}$ | 1.0 |
| Format Consistency ($R_{\mathrm{fmt}}$) | $\alpha$ | 1.0 |
| Region Selection ($R_{\mathrm{sel}}$) | $\beta_{\mathrm{pos}}$ | 1.0 |
| | $\beta_{\mathrm{neg}}$ | 0.6 |
| | $\gamma_{\varnothing}$ | 0.5 |
| Composite Reward | $(\lambda_1, \lambda_2)$ | (0.3, 0.5) |

## I   LIMITATIONS

While *DeFacto* demonstrates consistent improvements in answer accuracy and reasoning faithfulness, there are a few limitations to note. First, our current implementation relies on publicly available detectors (e.g., RPN, OCR, and open-vocabulary models) for region proposal, which may introduce occasional errors or inefficiencies; however, this can be alleviated as stronger detectors become available. Second, our counterfactual dataset consists about 100k images, which is sufficient for controlled experiments but still modest compared to large-scale pretraining corpora. Lastly, our framework has so far been evaluated on static images, leaving the extension to videos and temporal reasoning as an open direction for future work.

## J   BROADER IMPACT

Our work promotes safer and more interpretable multimodal reasoning by ensuring that models align their predictions with visual evidence. Beyond algorithmic contributions, we release a large-scale counterfactual dataset of about 100k images, which we believe will be a valuable resource for the community to study faithful reasoning.

## K   THE USE OF LARGE LANGUAGE MODELS (LLMS)

In this work, we employed LLMs in a limited capacity to support writing and presentation. Specifically, we used an LLM to help with linguistic refinement in the *Introduction* and *Related Work* sections, ensuring clarity and fluency of exposition. In addition, we used LLM assistance for formatting tasks in the *Method* and *Experiment* sections, such as rendering mathematical formulas into standard LATEX notation and typesetting tables in the appropriate style. All core research contributions, including algorithm design, dataset construction, experimental execution, and analysis, were entirely conducted by the authors without LLM involvement.

## L   CONCLUSION

In this work, we introduced *DeFacto*, the first vision-language reasoning framework explicitly grounded in counterfactual supervision, designed to enforce region-faithful reasoning and abstention behavior when critical evidence is missing. To enable this counterfactual reasoning paradigm, we proposed an automatic pipeline for constructing counterfactual datasets, which leverages language model parsing, open-vocabulary detection, and OCR to mask question-relevant regions without requiring manual annotations. Using this pipeline, we built a counterfactual dataset about 100k images to support training and evaluation. Extensive experiments across multiple diverse benchmarks

demonstrate that DeFacto consistently improves both answer accuracy and visual grounding faithfulness over strong baselines. Our ablation studies further confirm the necessity of counterfactual training and region-level reward design in enhancing interpretability and robustness. We believe these findings open new directions for integrating counterfactual supervision into multimodal reasoning systems, with potential extensions to video understanding and embodied AI.

