# OpenReview forum: "DeFacto: Counterfactual Thinking with Images for Enforcing Evidence-Grounded and Faithful Reasoning"
_ICLR.cc/2026/Conference — ICLR 2026 Conference Withdrawn Submission_

### Official Review · Reviewer_JgWv · 2025-10-23

**Soundness:** 3
**Presentation:** 4
**Contribution:** 3
**Rating:** 8
**Confidence:** 4

**Summary:**

This paper introduces a counterfactual reasoning framework aimed at improving the faithfulness and interpretability of multimodal language models. It ensures that models not only generate correct answers but also ground their reasoning on the relevant visual evidence. The proposed approach leverages Counterfactual GRPO training with three complementary paradigms (positive, counterfactual, and random-masking) to jointly enforce answer accuracy and reasoning consistency. In addition, the authors construct a large-scale counterfactual dataset using open-vocabulary detection and OCR-guided masking to automatically identify question-relevant regions. Extensive experiments across diverse benchmarks demonstrate significant improvements in both answer accuracy and reasoning faithfulness compared to strong baselines.

**Strengths:**

Pros:
1. The paper proposes a counterfactual training framework that leverages GRPO-based reinforcement learning to jointly optimize both answer correctness and region-level reasoning faithfulness.
2. The authors introduce an automated counterfactual dataset construction pipeline that effectively generates visual grounding cues (bounding boxes) to support multimodal large language model (MLLM) reasoning.
3. The experiments are comprehensive and well-structured, covering a wide range of benchmarks (e.g., VQAv2, OKVQA, GQA, DocVQA, TextVQA). The results demonstrate consistent and substantial improvements in both accuracy and reasoning faithfulness over strong baselines.
4. The paper is clearly written and well-organized, with structured explanations and illustrative figures that make the approach easy to understand and follow.

**Weaknesses:**

Cons:
1. The automatic evidence localization pipeline relies heavily on open-vocabulary detectors and OCR, which may introduce noise or misalignment when identifying evidence regions. In essence, the framework distills knowledge from these components, making it difficult for the model’s visual grounding capability to surpass that of the underlying detectors.
2. Accurately detecting bounding boxes in figure-type images (e.g., those in Figures 5, 11, and 17) is more challenging than in natural images, such as when distinguishing a specific polyline among multiple overlapping ones. This limitation likely contributes to the smaller performance gains observed in Table 2 compared to Table 1.

**Questions:**

Questions:
1. Could the authors provide quantitative metrics or human validation to assess how accurately the automatic pipeline identifies question-relevant evidence regions (R⁺)? For example, including human evaluation results on a small subset (e.g., 1,000 images) would help validate the reliability of the evidence localization process. Besides, show some failure examples in Appendix could also help us better understanding the datasets.
2. If the open-vocabulary detectors were used directly as external tools to validate all bounding boxes, and the positive filtered bounding boxes were then provided to the original Qwen2.5-VL-7B model in prompt, would the model achieve performance comparable to the GRPO-finetuned results?
3. The paper mentions a 100k-image dataset constructed from multiple benchmarks. Could the authors elaborate on the domain distribution (e.g., natural images, documents, charts) and discuss how domain balance may influence performance generalization?

---

> ### Author Response · Authors · 2025-11-22
>
> Thank you for your positive assessment of our work. Below we address the raised weaknesses.
>
>
> **Cons 1:**
> “The automatic evidence localization pipeline relies heavily on open-vocabulary detectors and OCR, which may introduce noise or misalignment when identifying evidence regions. In essence, the framework distills knowledge from these components, making it difficult for the model’s visual grounding capability to surpass that of the underlying detectors.”
>
> **Answer 1:**
> Thank you for the positive feedback on our work and for raising this concern. We agree that the automatic evidence localization pipeline may introduce noise or misalignment due to imperfect open-vocabulary detection and OCR. Such noise is indeed possible, but in practice it remains within a tolerable range—the majority of evidence regions are still correctly identified, allowing the constructed positive and counterfactual samples to preserve the intended supervision structure. To verify this, we conducted two experiments. **Table A** reports results on our 1k manually annotated faithfulness validation set, showing that DeFacto improves both grounding quality and answer accuracy compared with GRPO. **Table B** compares GRPO with and without counterfactual rewards across six benchmarks and again shows consistent improvements, demonstrating that moderate upstream noise does not hinder the effectiveness of our framework.
>
> **Table A: GRPO vs. DeFacto on the 1k Human-Annotated Faithfulness Set**
>
> | Model             | mAP   | AP50 | AP75 | Acc   |
> | ----------------- | ----- | ---- | ---- | ----- |
> | Qwen2.5-VL + GRPO | 20.4  | 28.8 | 18.9 | 65.0  |
> | **DeFacto (ours)** | **30.6** | **36.1** | **24.8** | **79.4** |
> | Δ                 | +10.2 | +7.3 | +5.9 | +14.4 |
>
> **Table B: GRPO With vs. Without Counterfactual (CF) Reward**
>
> | Model Variant                   | VQAv2 | OKVQA | SciQA | VSR   | DocVQA | TextVQA |
> | ------------------------------- | ----- | ----- | ----- | ----- | ------ | ------- |
> | Qwen2.5-VL (GRPO, no CF reward) | 70.4  | 56.9  | 85.9  | 58.4  | 85.4   | 72.8    |
> | **DeFacto (CF reward + GRPO)** | **79.7** | **68.0** | **88.2** | **70.3** | **85.8** | **73.4** |
> | Δ                               | +9.3  | +11.1 | +2.3  | +11.9 | +0.4   | +0.6    |
>
> Regarding the concern that our framework may simply distill the knowledge of upstream detectors and therefore be limited by them, we clarify that these components are not treated as teachers that define the model’s grounding capability. Their role is limited to supplying initial candidate regions from which we construct evidence-present and evidence-masked samples. While these candidates influence the sampling process, the actual supervision comes from whether the model can correctly answer under counterfactual masking, rather than from matching detector outputs. The reward signal is thus driven by the model’s own reasoning consistency, not by imitation of the upstream detectors. Because the model aggregates signals across many diverse samples and is not required to reproduce detector boxes, it can—and empirically does—learn grounding behavior that exceeds the quality implied by the initial proposals, as illustrated in Table A.
>
> All related clarifications and the supporting experimental results have been added to Section 4.2 (Faithful reasoning evaluation) and Section 4.3 (Ablation Studies) in the revised manuscript.
>
>
> **Cons 2:**
> “Accurately detecting bounding boxes in figure-type images (e.g., those in Figures 5, 11, and 17) is more challenging than in natural images, such as when distinguishing a specific polyline among multiple overlapping ones. This limitation likely contributes to the smaller performance gains observed in Table 2 compared to Table 1.”
>
> **Answer 2:**
> Thank you for the insightful observation. Figure-type images such as charts and diagrams indeed present different visual characteristics compared with natural images, often containing overlapping lines, dense symbolic elements, or abstract graphical structures. Our current pipeline uses general-purpose open-vocabulary detection and OCR, which are not specifically optimized for these structured figures. As a result, evidence localization in such scenarios can be less precise, which may partially explain the relatively smaller performance gains observed compared with natural-image benchmarks.
>
> That said, DeFacto still provides consistent improvements even under these challenging settings, indicating that its counterfactual supervision remains beneficial despite imperfect region proposals. We consider enhancing figure-type understanding—such as through structure-aware detection, relation parsing, or chart-specific representations—an important direction for future work.

---

> ### Author Response · Authors · 2025-11-22
>
> **Question 1:**
> “Could the authors provide quantitative metrics or human validation to assess how accurately the automatic pipeline identifies question-relevant evidence regions (R⁺)? For example, including human evaluation results on a small subset (e.g., 1,000 images) would help validate the reliability of the evidence localization process. Besides, showing some failure examples in the Appendix could also help us better understand the datasets.”
>
> **Answer 1:**
> Thank you for the suggestion. This is an important point for evaluating how well the pipeline identifies question-relevant evidence. To assess this, we constructed a manually annotated 1,000-image validation set. For each example, annotators labeled the ground-truth evidence region corresponding to the correct answer. Accordingly, mAP, AP50, and AP75 quantify the joint correctness of evidence localization and answer prediction. We additionally report answer accuracy on the same set. The quantitative results are shown in Table C.
>
> **Table C: Human-Annotated 1k Evidence-Localization Evaluation**
>
> | Model             | mAP  | AP50 | AP75 | Accuracy (%) |
> | ----------------- | ---- | ---- | ---- | ------------ |
> | GRIT              | 0.0  | 0.0  | 0.0  | 73.7         |
> | DeepEyes          | 0.0  | 2.2  | 0.9  | 44.0         |
> | Qwen2.5-VL + GRPO | 20.4 | 28.8 | 18.9 | 65.0         |
> | **DeFacto (ours)** | **30.6** | **36.1** | **24.8** | **79.4** |
>
> These results show that DeFacto produces significantly more faithful evidence regions than existing baselines.
>
> In addition, we have included representative failure cases of the evidence-localization process in Section 5 (Visualization and Error Analysis), illustrating common issues such as partially captured evidence or missed text segments.
>
> All new quantitative evaluations and qualitative examples have been added to **Section 4.2** (Faithful reasoning evaluation) and **Section 5** of the revised manuscript.
>
>
> **Question 2:**
> “If the open-vocabulary detectors were used directly as external tools to validate all bounding boxes, and the positive filtered bounding boxes were then provided to the original Qwen2.5-VL-7B model in prompt, would the model achieve performance comparable to the GRPO-finetuned results?”
>
> **Answer 2:**
> Thank you for the thoughtful question. To examine this possibility, we also evaluated the alternative strategy of directly supplying detector-generated positive bounding boxes (from open-vocabulary detection and OCR) to the original Qwen2.5-VL-7B model without any GRPO finetuning. In this setup, the model receives the filtered boxes in the prompt exactly as an external tool would provide them. As shown in Table D, this approach results in noticeably lower performance across all benchmarks compared with DeFacto.
>
> **Table D: OVD/OCR + Qwen2.5-VL vs. DeFacto**
>
> | Setting              | OKVQA | GQA   | ChartQA | TextVQA | AI2D  |
> | -------------------- | ----- | ----- | ------- | ------- | ----- |
> | OVD/OCR + Qwen2.5-VL | 44.7  | 52.1  | 65.0    | 66.8    | 61.8  |
> | **DeFacto (ours)** | **68.0** | **70.1** | **82.4** | **73.4** | **79.0** |
> | Δ                    | +23.3 | +18.0 | +17.4   | +6.6    | +17.2 |
>
> These results suggest that simply prompting the model with detector-filtered boxes is not sufficient to achieve strong performance. The key benefits of DeFacto come from the counterfactual rewards and region-level constraints introduced during GRPO training, which encourage the model to use evidence consistently, refrain from answering when evidence is missing, and avoid shortcut behaviors rather than merely consuming detector outputs.
>
>
> **Question 3:**
> “The paper mentions a 100k-image dataset constructed from multiple benchmarks. Could the authors elaborate on the domain distribution (e.g., natural images, documents, charts) and discuss how domain balance may influence performance generalization?”
>
> **Answer 3:**
> Thank you for the question. To provide a clearer view of the domain composition of the 100k-image dataset, we added a new section in the revised manuscript—Appendix B (“Dataset Distribution Visualization”). This appendix summarizes the distribution across the major visual domains represented in our data. Specifically, the dataset is grouped into three categories:
>
> * General VQA: 47.67%
> * Scene text–centric VQA: 22.05%
> * Document-oriented VQA: 30.28%
>
> This distribution ensures that the training data covers both natural-image scenarios and text-heavy document or chart-like images. Such domain diversity helps prevent overfitting to a single modality and supports better generalization across heterogeneous visual reasoning tasks, particularly when the model is required to integrate both perception and text understanding.
>
> All clarifications and visualizations have been added to **Appendix B** in the revised manuscript.

---

> ### Comment · Reviewer_JgWv · 2025-11-26
> **Official Comment by Reviewer JgWv**
>
> In my initial assessment, I found the paper’s motivation and results are very convincing. Specifically, employing Counterfactual Thinking to guide the model to attend to the correct grounding and thus enhancing overall performance makes sense for me. However, after reviewing the discussion between the authors and Reviewer oCdo, I find myself become confused.
>
> The authors state that 'Standard GRPO produces more faithful grounding, though this does not necessarily yield higher accuracy.' Does this imply that faithful grounding is not the primary cause of improved model accuracy? Furthermore, Table C demonstrates that GRIT's accuracy is not positively correlated with its grounding performance. If this is the case, what is the actual source of the performance gains attributed to the proposed Counterfactual Thinking?
>
> In light of these concerns, I have decided to lower my rating to 6. Nevertheless, I still consider the research direction of this paper to be highly valuable. However, the authors need to revisit and better explain the phenomena observed in Table C, as the current discussion with Reviewer oCdo has exacerbated my confusion rather than resolving it.

---

> ### Author Response · Authors · 2025-11-26
>
> Thank you for the follow-up. To clarify, the statement “though this does not necessarily yield higher accuracy” refers specifically to standard GRPO and not all reasoning paradigms. While GRPO produces more faithful grounding, it does not learn to generate correct answers based on that evidence, as there is no consistency constraint between the reasoning process and the final answer. This results in high evidence faithfulness, but the answers are still not optimal because the reasoning and the answer are not aligned. In other words, GRPO's grounding is faithful, but the model does not learn to use it to improve accuracy, which is why its overall performance is limited despite better grounding.
>
> Regarding the observation that Table C shows GRIT has high accuracy despite poor grounding, this also fits within our motivation. GRIT is trained with a high weight on answer correctness, which encourages it to learn a shortcut: as long as the answer is correct, the predicted evidence does not matter. This leads the model to ignore grounding entirely, producing high accuracy but unfaithful reasoning. Thus, GRIT’s behavior is not evidence against our method, but an illustration of why enforcing reasoning–answer alignment is necessary.
>
> As stated explicitly in lines 92–98 of the manuscript, “correct answers alone are not sufficient—the reasoning process itself must also be correct... What is needed is a training paradigm that enforces both correct evidence selection and correct answering, ensuring that reasoning trajectories and final predictions are jointly faithful to the visual input.” Faithful reasoning, rather than accuracy alone, is the motivation and goal of our work.
>
> To address the concern about whether the reasoning process is truly effective, we designed an additional experiment to verify the consistency between reasoning and answers. In this experiment, all variants were trained using the same backbone, training data, and hyperparameters, with the only difference being whether counterfactual supervision was applied. This allows us to fairly compare the impact of counterfactual supervision on reasoning consistency. We compare the following variants:
>
> 1. Qwen2.5-VL (base model)
> 2. Qwen2.5-VL + SFT (no CF)
> 3. Qwen2.5-VL + SFT (CF alignment)
> 4. Qwen2.5-VL + GRPO (no CF reward)
> 5. DeFacto (our counterfactual method)
>
> We evaluated the models using the same 1k-annotated validation set for evidence grounding and answer accuracy, with all models tested using the same prompt.
>
> **Table F Faithfulness Evaluation**
>
> | Model Variant                    | mAP  | AP50 | AP75 | Accuracy (%) |
> | -------------------------------- | ---- | ---- | ---- | ------------ |
> | Qwen2.5-VL (Base)                | 2.3 | 1.2 | 1.5 | 55.1         |
> | Qwen2.5-VL + SFT (no CF)         | 15.7 | 13.9 | 12.8 | 61.4         |
> | Qwen2.5-VL + SFT (CF alignment)  | 18.9 | 16.4 | 15.7 | 63.8        |
> | Qwen2.5-VL + GRPO (no CF reward) | 20.4 | 28.8 | 18.9 | 65.0         |
> | **DeFacto (ours)**|**30.6**|**36.1**|**24.8**|**79.4**|
>
> These results demonstrate that counterfactual supervision improves both reasoning–answer consistency and accuracy, and the gains are not just from better grounding. The controlled experiment shows that the improvements come from the joint enforcement of correct reasoning and correct answers, which is the core motivation of our work.
>
> We will revise the manuscript to further clarify this relationship and avoid any confusion.

---

### Official Review · Reviewer_o15t · 2025-10-24

**Soundness:** 2
**Presentation:** 2
**Contribution:** 2
**Rating:** 4
**Confidence:** 3

**Summary:**

“Thinking with images” can lead to correct answers even when the reasoning process is irrelevant. The paper proposes a framework called DeFacto, a counterfactual reasoning framework that aligns reasoning trajectories with supporting evidence. The key idea is to jointly optimize the model with GRPO, not only for accuracy and format reward, but also through (1) positive supervision, (2) counterfactual abstention, and (3) random masking. To enable this optimization, a new counterfactual dataset is automatically constructed. After counterfactual training, the models perform significantly better compared with the based models.

**Strengths:**

1. The proposed approach is simple yet effective, and the paper is easy to follow.

2. A new counterfactual dataset is introduced, which could benefit the research community.

3. The improvement of the base model after training with the counterfactual dataset is significant.

**Weaknesses:**

1. The paper lacks comparisons with several recent works listed in [1].

2. The correctness of the constructed dataset depends on multiple upstream models.

3. The benchmarks presented in the main paper are not recent. Although some recent benchmarks are included in the appendix, the selected models differ from those listed in the main paper.

Reference:
[1] Thinking with Images for Multimodal Reasoning: Foundations, Methods, and Future Frontiers

**Questions:**

What is the model’s performance on more recent benchmarks?

How does it compare with the recent models mentioned in [1]?

Reference:
[1] Thinking with Images for Multimodal Reasoning: Foundations, Methods, and Future Frontiers

---

> ### Author Response · Authors · 2025-11-22
>
> **Weakness 1:**
> “The paper lacks comparisons with several recent works listed in [1]. Reference: [1] *Thinking with Images for Multimodal Reasoning: Foundations, Methods, and Future Frontiers*.”
>
> **Answer 1:**
> Thank you for the suggestion. The works surveyed in [1] span multiple categories within the “thinking with images’’ landscape, and our method falls under the RL-based visual reasoning family. To address this concern, we have added comparisons with two representative RL-based methods highlighted in [1]: *Chain-of-Focus* [2] and *Pixel Reasoner* [3]. These methods similarly employ reinforcement learning to guide region-level or pixel-level visual interactions, making them the most relevant baselines for evaluating DeFacto.
>
> We report the results in Table A below. These comparisons have been fully incorporated into the revised manuscript, specifically in **Appendix A (Additional Results)** under the *Thinking with Images* part in  Table 5.
>
> **Table A: Comparison with RL-based “Thinking with Images’’ Methods**
>
> | Model             | OCRBench | MMstar   | MMMU     | MMB1.1   | POPE     |
> | ----------------- | -------- | -------- | -------- | -------- | -------- |
> | Chain-of-Focus    | 632      | 58.1     | 46.1     | 75.3     | 88.4     |
> | Pixel Reasoner    | 597      | 62.9     | 52.5     | 78.5     | 87.8     |
> |**DeFacto-7B (ours)** | **871**  | **63.2** | **56.6** | **81.2** | **88.6** |
>
> Across all five benchmarks, DeFacto achieves the strongest performance, with the most substantial gains appearing on **OCRBench**, **MMMU**, and **MMB1.1**—demonstrating its advantages in text-intensive reasoning as well as fine-grained multimodal understanding.
>
> All related comparisons and discussions have been integrated into the revised manuscript.
>
>
>
> **Weakness 2:**
> “The correctness of the constructed dataset depends on multiple upstream models.”
>
> **Answer 2:**
> Thank you for the comment. While the dataset construction pipeline relies on region proposals, OCR, and open-vocabulary detection, it is indeed possible for these upstream components to occasionally mis-detect or mis-localize evidence. However, such errors remain moderate in scale and do not undermine the overall correctness of the constructed dataset. Since DeFacto relies on distinguishing whether essential evidence is available or masked rather than on exact detector outputs, the supervision signal remains reliable. In practice, occasional detection mistakes introduce tolerable noise, but the majority of evidence regions are still correctly identified, allowing the model to learn stable and consistent counterfactual behavior.
>
> To empirically confirm this robustness, **Table B** evaluates grounding quality and answer accuracy on a 1k manually annotated validation set, showing that DeFacto substantially outperforms GRPO. **Table C** reports a broader comparison across six benchmarks, where DeFacto again achieves consistent gains, indicating that moderate noise in the constructed dataset does not hinder the method’s effectiveness.
>
> **Table B: GRPO vs. DeFacto on the 1k Human-Annotated Faithfulness Set**
>
> | Model             | mAP      | AP50     | AP75     | Acc      |
> | ----------------- | -------- | -------- | -------- | -------- |
> | Qwen2.5-VL + GRPO | 20.4     | 28.8     | 18.9     | 65.0     |
> | **DeFacto-7B (ours)**| **30.6** | **36.1** | **24.8** | **79.4** |
> | Δ                 | +10.2    | +7.3     | +5.9     | +14.4    |
>
> **Table C: GRPO With vs. Without Counterfactual (CF) Reward**
>
> | Model Variant                   | VQAv2    | OKVQA    | SciQA    | VSR      | DocVQA   | TextVQA  |
> | ------------------------------- | -------- | -------- | -------- | -------- | -------- | -------- |
> | Qwen2.5-VL (GRPO, no CF reward) | 70.4     | 56.9     | 85.9     | 58.4     | 85.4     | 72.8     |
> | **DeFacto (CF reward + GRPO) **| **79.7** | **68.0** | **88.2** | **70.3** | **85.8** | **73.4** |
> | Δ                               | +9.3     | +11.1    | +2.3     | +11.9    | +0.4     | +0.6     |
>
> **Table B** has been added to **Section 4.2 (Faithful Reasoning Evaluation)** of the revised manuscript, and **Table C** appears in **Section 4.3 (Ablation Studies)** for detailed inspection.

---

> ### Author Response · Authors · 2025-11-22
>
> **Weakness 3:**
> “The benchmarks presented in the main paper are not recent. Although some recent benchmarks are included in the appendix, the selected models differ from those listed in the main paper.”
>
>
>
> **Answer 3:**
> Thank you for the suggestion. In the revised version, we expanded the evaluation to ensure that all experiments use a consistent set of baselines and that recent benchmarks are thoroughly covered.
>
> 1. We added the POPE benchmark and re-evaluated all baselines used in the main paper, together with the newly added RL-based methods from [1]—Chain-of-Focus [2] and Pixel-Reasoner [3].
>    DeFacto achieves the strongest performance across all five recent benchmarks (OCRBench, MMStar, MMMU, MMB1.1, POPE), showing its effectiveness on both perception-heavy and text-rich tasks.
>
> 2. We included results on our 1k manually annotated reasoning-faithfulness validation set, which jointly evaluates evidence localization and answer correctness.
>    On this dataset, DeFacto significantly improves grounding quality (mAP/AP50/AP75) and overall accuracy compared with GRIT, DeepEyes, and GRPO, demonstrating that its reasoning traces are more faithful and better aligned with human-annotated evidence.
>
> 3. We added evaluation on the MSTI 2.0 dataset, which measures object detection, entity recognition, and answer generation in a unified setting.
>    DeFacto again outperforms both zero-shot Qwen2.5-VL and GRPO-CoT variants on all MSTI metrics (EM, F1, AP), confirming that its counterfactual training also benefits structured visual understanding tasks.
>
> The updated tables are included below.
>
> ---
>
> **Table D: Evaluation on Recent Benchmarks**
>
> | Model                 | OCRBench | MMstar   | MMMU     | MMB1.1   | POPE     |
> | --------------------- | -------- | -------- | -------- | -------- | -------- |
> | GRIT                  | 322      | 36.3     | 17.1     | 9.7      | 85.7     |
> | ViCrop                | 233      | 33.1     | 26.1     | 51.7     | 87.3     |
> | DeepEyes              | 636      | 43.6     | 44.1     | 29.4     | 87.7     |
> | Visual-SR1            | 449      | 62.8     | 57.2     | 77.4     | 86.0     |
> | Chain-of-Focus        | 632      | 58.1     | 46.1     | 75.3     | 88.4     |
> | Pixel Reasoner        | 597      | 62.9     | 52.5     | 78.5     | 87.8     |
> | **DeFacto-7B (ours)** | **871**  | **63.2** | **56.6** | **81.2** | **88.6** |
>
> ---
>
> **Table E: 1k Manually Annotated Faithfulness Set**
>
> | Model              | mAP      | AP50     | AP75     | Accuracy (%) |
> | ------------------ | -------- | -------- | -------- | ------------ |
> | GRIT               | 0.0      | 0.0      | 0.0      | 73.7         |
> | DeepEyes           | 0.0      | 2.2      | 0.9      | 44.0         |
> | Qwen2.5-VL + GRPO  | 20.4     | 28.8     | 18.9     | 65.0         |
> | **DeFacto (ours)** | **30.6** | **36.1** | **24.8** | **79.4**     |
>
> ---
>
> **Table F: Results on the MSTI 2.0 Dataset**
>
> | Model                  | Dev EM    | Dev F1   | Dev AP   | Test EM   | Test F1  | Test AP  |
> | ---------------------- | --------- | -------- | -------- | --------- | -------- | -------- |
> | Qwen2.5-VL (Zero-shot) | 22.2     | 1.1     | 1.2     | 18.6    | 2.6     | 0.7     |
> | Qwen2.5-VL (GRPO-CoT)  | 24.3     | 1.4    | 3.3     | 19.7     | 3.1     | 1.2     |
> | **DeFacto**            | **28.4** | **1.6** | **4.1** | **23.1** | **4.5** | **2.5** |
>
> ---
>
> **Table D** has been added to **Appendix A (Table 5)**.
> **Table E** appears in **Section 4.2 (Faithful reasoning evaluation)**.
> **Table F** has been added to **Appendix A (Table 6)**.
>
>
> **Question 1:**
> "What is the model’s performance on more recent benchmarks?"
>
> **Answer 1:**
> Thank you for the question. Our updated evaluation on recent benchmarks is presented in the response to **Weakness 3**, where we substantially expanded the experimental coverage. Specifically, we newly evaluated all “thinking with images’’ baselines—including GRIT, DeepEyes, ViCrop, Visual-SR1, Chain-of-Focus, and Pixel-Reasoner—on five recent benchmarks (OCRBench, MMStar, MMMU, MMB1.1, POPE). In addition, we incorporated results from our **1k manually annotated reasoning-faithfulness validation set**, and we further added evaluation on the **MSTI 2.0** dataset, which tests object detection, entity recognition, and answer generation. The updated results, summarized in Table D below, show that DeFacto achieves the strongest performance across all recent benchmarks.
>
>
> **Question 2:**
> "How does it compare with the recent models mentioned in [1]?"
>
> **Answer 2:**
> Thank you for the question. As discussed in our response to **Weakness 1**, we incorporated two representative recent RL-based “thinking with images’’ methods from [1]—**Chain-of-Focus** and **Pixel Reasoner**—into our evaluation. These models are compared alongside all baselines on the recent benchmarks, and DeFacto consistently outperforms both methods across OCRBench, MMStar, MMMU, MMB1.1, and POPE.

---

### Official Review · Reviewer_8FmZ · 2025-10-29

**Soundness:** 3
**Presentation:** 3
**Contribution:** 3
**Rating:** 4
**Confidence:** 4

**Summary:**

This paper introduces ​​DeFacto​​, a novel counterfactual reasoning framework designed to enhance the evidence-groundedness and faithfulness of multimodal large language models (MLLMs). The core innovation addresses a critical weakness in existing "thinking with images" paradigms: models often arrive at correct answers by relying on spurious or irrelevant image regions, indicating a disconnect between their reasoning and the actual visual evidence. To solve this, DeFacto employs three complementary training paradigms, ​​positive, counterfactual, and random-masking​​, which are automatically constructed using a pipeline that localizes question-relevant evidence. The models are then trained using GRPO-based reinforcement learning with a composite reward function that jointly enforces ​​answer correctness, format consistency, and region selection coherence​​.

**Strengths:**

- The work introduces an innovative approach that effectively addresses a key limitation of previous methods in the field.
- It demonstrates significant and consistent improvements over state-of-the-art baselines across multiple benchmark datasets.
- The proposed framework is efficient and has the potential for broad application in real-world scenarios.

**Weaknesses:**

- DeFacto's data construction process relies heavily on components such as region proposal networks and open vocabulary detectors. The performance limitations of these upstream models (such as missed detections and false detections) will directly introduce noise, affecting the quality of counterfactual samples, thereby limiting its performance ceiling.
- Lack of the necessary experiments on the suitable task MSTI [a], which includes the object detection, entity recognition and answer generation.
- Lack of the necessary case studies and error analysis to intuitively illustrate the success and failure modes of the proposed DeFacto work.

[a] Chen, Zixin, et al. "CofiPara: A coarse-to-fine paradigm for multimodal sarcasm target identification with large multimodal models." Proceedings of the 62nd Annual Meeting of the Association for Computational Linguistics (Volume 1: Long Papers). 2024.

**Questions:**

How to evaluate the quality of the <think> part in terms of explainability or helpfulness or coherence?

---

> ### Author Response · Authors · 2025-11-22
>
> Thank you for your recognition of the novelty of our method. Below we address your concerns.
>
>
> **Weakness 1:**
> “DeFacto's data construction process relies heavily on components such as region proposal networks and open-vocabulary detectors. The performance limitations of these upstream models may introduce noise that affects the quality of counterfactual samples, potentially limiting the performance ceiling.”
>
> **Answer 1:**
> Thank you for the suggestion. To directly assess the effect of upstream detection noise, we constructed a manually annotated 1,000-image validation set that evaluates both evidence accuracy and answer accuracy. We use mAP, AP50, and AP75 to measure how well the predicted regions overlap with human-annotated evidence, and answer accuracy to measure correctness. The results, summarized in Table A, show clear and consistent improvements over GRPO, indicating that DeFacto remains robust even when upstream detections are imperfect.
>
>  **Table A: Qwen2.5-VL + GRPO vs. DeFacto (1k Human-Annotated Validation Set)**
>
> | Model                  | mAP    | AP50   | AP75   | Acc    |
> | ---------------------- | ------ | ------ | ------ | ------ |
> | Qwen2.5-VL + GRPO      | 20.4   | 28.8   | 18.9   | 65.0   |
> | **DeFacto (ours)**         | **30.6** | **36.1** | **24.8** | **79.4** |
> | $\Delta$               | +10.2  | +7.3  | +5.9  | +14.4  |
>
> We further include an ablation in Section 4.3 comparing standard GRPO trained on the original data with DeFacto trained using our annotated evidence and counterfactual rewards. Table B shows consistent gains across six benchmarks.
>
>  **Table B: GRPO w/ and w/o Counterfactual (CF) Reward**
>
> | Model Variant                   | VQAv2   | OKVQA   | SciQA   | VSR     | DocVQA  | TextVQA |
> | ------------------------------- | ------- | ------- | ------- | ------- | ------- | ------- |
> | Qwen2.5-VL (GRPO, no CF reward) | 70.4    | 56.9    | 85.9    | 58.4    | 85.4    | 72.8    |
> | **DeFacto (CF reward + GRPO)**      | **79.7** | **68.0** | **88.2** | **70.3** | **85.8** | **73.4** |
> | $\Delta$                        | +9.3    | +11.1   | +2.3    | +11.9   | +0.4    | +0.6    |
>
> Taken together, Table A and Table B demonstrate that moderate detection noise does not limit DeFacto’s effectiveness;
>
> All related clarifications and results have been incorporated into the revised manuscript, specifically in Section 4.2 (*Faithful reasoning evaluation*) and Section 4.3 (*Ablation Studies*).
>
>
>
>
> **Weakness 2:**  "Lack of the necessary experiments on the suitable task MSTI [a], which includes the object detection, entity recognition and answer generation."
>
>
> **Answer 2:**
> Thank you for suggesting MSTI 2.0 as an additional benchmark. Following your advice, we have expanded our experiments to include this dataset, which evaluates three key capabilities: object detection, entity recognition, and answer generation. Accordingly, we report EM for answer correctness, F1 for entity recognition, and AP for object-level localization. In our evaluation, we conduct experiments on both the MSTI 2.0 development split (dev) and test split (test), allowing us to assess generalization and robustness across settings. The results, presented in Table C, show that DeFacto achieves consistent improvements over both zero-shot Qwen2.5-VL and the GRPO-CoT variant.
>
> **Table C: Performance of Qwen2.5-VL Variants and DeFacto on MSTI 2.0**
>
> | Model                  | Dev EM    | Dev F1   | Dev AP   | Test EM   | Test F1  | Test AP  |
> | ---------------------- | --------- | -------- | -------- | --------- | -------- | -------- |
> | Qwen2.5-VL (Zero-shot) | 22.2     | 1.1     | 1.2     | 18.6    | 2.6     | 0.7     |
> | Qwen2.5-VL (GRPO-CoT)  | 24.3     | 1.4    | 3.3     | 19.7     | 3.1     | 1.2     |
> | **DeFacto**            | **28.4** | **1.6** | **4.1** | **23.1** | **4.5** | **2.5** |
>
> We have incorporated the full results and discussion into Appendix A and updated the main PDF accordingly. The reference you provided has also been added and discussed in the related work section.
>
>
>
>
> **Weakness 3:** “Lack of the necessary case studies and error analysis to intuitively illustrate the success and failure modes of the proposed DeFacto work.”
>
> **Answer 3:** Thank you for the suggestion. We have added additional case studies and error analysis in Section 5, and visualized several representative failure examples in Figure 3. The analysis summarizes four main failure types of DeFacto:
>  (1) semantic ambiguity in spatial expressions
>  (2) subjective or vague attribute definitions
>  (3) ambiguous comparative reasoning
>  (4) fine-grained human action confusion. These examples help clarify the current limitations and indicate directions for improving semantic disambiguation and fine-grained perception.
>
> We also added visualization comparisons between DeFacto and GRPO in Appendix D (Figure 21-Figure 28), illustrating both successful cases and how DeFacto selects more relevant evidence regions than GRPO.

---

> ### Author Response · Authors · 2025-11-23
>
> **Question 1**: "How to evaluate the quality of the [object Object] part in terms of explainability or helpfulness or coherence?"
>
> **Answer:**
> Thank you for the question. To evaluate the explainability, helpfulness, and coherence of the model-generated *[object Object]* component, we constructed a manually annotated 1,000-image validation set. This set allows us to explicitly measure whether the predicted visual regions are aligned with the human-identified evidence that supports the answer.
>
> We use mAP, AP50, and AP75 to evaluate the joint correctness of the predicted region and the predicted answer. These metrics therefore quantify how well the model identifies the evidence that actually supports its answer. We additionally report answer accuracy to measure overall prediction quality. The quantitative results are summarized in Table C.
>
> **Table D: Human-Annotated 1k Evidence-Localization Evaluation**
>
> | Model             | mAP  | AP50 | AP75 | Accuracy (%) |
> | ----------------- | ---- | ---- | ---- | ------------ |
> | GRIT              | 0.0  | 0.0  | 0.0  | 73.7         |
> | DeepEyes          | 0.0  | 2.2  | 0.9  | 44.0         |
> | Qwen2.5-VL + GRPO | 20.4 | 28.8 | 18.9 | 65.0         |
> | **DeFacto (ours)** | **30.6** | **36.1** | **24.8** | **79.4** |
>
> We then added a new experiment to further validate our motivation, which emphasizes the importance of ensuring consistency between reasoning and answers.
>
> As shown in Table D, DeFacto outperforms all other variants in both grounding quality and accuracy, confirming that our approach improves both aspects simultaneously.
>
> **Table F: Faithfulness Evaluation**
>
>
> | Model Variant                    | mAP  | AP50 | AP75 | Accuracy (%) |
> | -------------------------------- | ---- | ---- | ---- | ------------ |
> | Qwen2.5-VL (Base)                | 2.3 | 1.2 | 1.5 | 55.1         |
> | Qwen2.5-VL + SFT (no CF)         | 15.7 | 13.9 | 12.8 | 61.4         |
> | Qwen2.5-VL + SFT (CF alignment)  | 18.9 | 16.4 | 15.7 | 63.8        |
> | Qwen2.5-VL + GRPO (no CF reward) | 20.4 | 28.8 | 18.9 | 65.0         |
> | **DeFacto (ours)**|**30.6**|**36.1**|**24.8**|**79.4**|

---

### Official Review · Reviewer_oCdo · 2025-10-31

**Soundness:** 3
**Presentation:** 2
**Contribution:** 4
**Rating:** 6
**Confidence:** 5

**Summary:**

This paper targets the problem that existing MLLMs often reason based on irrelevant visual regions. It identifies two failure types: Mislocalized Failure (attending to wrong regions) and Spurious Correctness (producing correct answers from unfaithful evidence). To address this, DeFacto introduces a GRPO-based training framework with counterfactual and random-masking paradigms, encouraging evidence-grounded reasoning and abstention when evidence is missing. The method improves accuracy on several VQA benchmarks.

**Strengths:**

1. The motivation is fundamental and important. Making the model’s reasoning truly grounded in visual evidence is a key step toward trustworthy multimodal reasoning, and this topic is still relatively underexplored.
2. The design of the region selection coherence reward and answer correctness reward is conceptually sound and practically reasonable.
3. The overall training idea is clear, and the results show noticeable quantitative improvements on several benchmarks.

**Weaknesses:**

[Major Weakness]
The main concern is that the paper does not directly verify whether DeFacto actually achieves its goal which is to enable the model’s reasoning to be faithful to the image.
1. Although Table 3 includes an ablation between DeFacto and GRPO, this only compares answer accuracy. It does not measure whether the reasoning process itself becomes more visually faithful. As a result, there should be quantitative results  showing whether the thinking process indeed correctly utilizes the predicted visual information (e.g., selected regions or bounding boxes) to answer after applying DeFacto.
2. The paper also does not analyze how often other methods (like GRIT or Visual-SR1) fail due to unfaithful reasoning or succeed despite unfaithful reasoning. Showing these ratios before and after using DeFacto would strongly support the claimed motivation.

[Minor Weakness]
1. It would be beneficial to evaluate DeFacto on additional model backbones (e.g., InternVL3.5) to assess its generalizability across architectures.
2. The paper lacks qualitative comparisons on the reasoning processes between DeFacto and GRPO.

In short, while the method shows numerical improvement, it remains unclear whether the model actually “thinks with the right regions”. These analyses are essential to confirm the motivation of this work, and my overall rating will depend heavily on how the authors address the points in major weakness.

**Questions:**

1. In Table 1, the term “CF reward” is not clearly defined. Based on my understanding, CF reward refers to your added components (the region selection coherence and answer correctness rewards) combined with the original GRPO training. Please confirm if this interpretation is correct.

---

> ### Author Response · Authors · 2025-11-19
>
> Thank you for your positive assessment of our work and for pointing out some weaknesses. We have updated the manuscript to address  concerns.
>
> **Weakness 1 (Major) :**
> “The main concern is that the paper does not directly verify whether DeFacto actually achieves its goal of enabling the model’s reasoning to be faithful to the image. Although Table 3 includes an ablation between DeFacto and GRPO, it only compares answer accuracy and does not evaluate whether the reasoning process itself becomes more visually faithful. There should be quantitative results showing whether the thinking process correctly utilizes the predicted visual information after applying DeFacto.”
>
> **Answer 1:**
> Thank you for pointing this out. To directly evaluate whether DeFacto improves the visual faithfulness of the reasoning process, we constructed a new 1k manually annotated validation set. This set was specifically designed to assess whether the model not only answers correctly but also grounds its reasoning in the correct visual regions. For each sample, human annotators label the evidence regions associated with the answer. During evaluation, the model’s predicted answer is treated as the category for computing mAP, AP50, and AP75: we compare the predicted evidence regions for that answer with the human-annotated evidence of the same answer category. This directly measures whether the model’s reasoning is consistent with the visual evidence required for its predicted answer.
>
> The results are shown in Table A.
>
> **Table A: Reasoning Faithfulness on the 1k Human-Annotated Validation Set**
>
> | Model             | mAP  | AP50 | AP75 | Accuracy (%) |
> | ----------------- | ---- | ---- | ---- | ------------ |
> | GRIT              | 0.0  | 0.0  | 0.0  | 73.7         |
> | DeepEyes          | 0.0  | 2.2  | 0.9  | 44.0         |
> | Qwen2.5-VL + GRPO | 20.4 | 28.8 | 18.9 | 65.0         |
> | **DeFacto (ours)** | **30.6** | **36.1** | **24.8** | **79.4** |
>
> These results provide several insights:
>
> 1. GRIT tends to rely on shortcuts and biases, leading to reasonable accuracy but nearly zero grounding alignment. The model often answers without referencing the correct evidence, revealing that the reasoning chain is not visually grounded.
> 2.DeepEyes  improves cropping behaviors but does not reliably identify which regions matter. It often zooms into irrelevant areas, so the reasoning remains weakly grounded despite the additional visual actions.
> 3.GRPO offers some improvements but still lacks explicit constraints that tie reasoning steps to the underlying image evidence, resulting in moderate but inconsistent grounding quality.
> 4.DeFacto achieves the strongest grounding fidelity. Its counterfactual training encourages the model to rely on the evidence that is actually present and to avoid unsupported reasoning. This results in reasoning traces that are more aligned with human-annotated evidence and more consistent with the intended learning objective.
>
> All new evaluations and analyses have been added to Section 4.2 (“Faithful reasoning evaluation”).
>
>
> **Weakness 2 (Major) :**
> “The paper also does not analyze how often other methods fail due to unfaithful reasoning or succeed despite unfaithful reasoning. Showing these ratios before and after using DeFacto would strongly support the claimed motivation.”
>
> **Answer 2:**
> Thank you for the suggestion. We have added a short analysis in Section 4.2 based on the new 1k human-annotated faithfulness set. The grounding metrics in Table A directly reveal how often each method’s predicted evidence fails to match the human-annotated regions. GRIT and DeepEyes show almost zero mAP/AP scores, indicating that even when they answer correctly, their selected regions rarely correspond to the true evidence. GRPO performs better but still shows substantial misalignment between predicted and annotated evidence. In contrast, DeFacto achieves markedly higher mAP, AP50, and AP75, demonstrating that it substantially reduces these unfaithful-reasoning cases. This comparison provides a clear quantification of how DeFacto improves the consistency between reasoning and visual evidence.
>
>
>
> **Weakness 3 (Minor) :**
> “It would be beneficial to evaluate DeFacto on additional model backbones (e.g., InternVL3.5) to assess its generalizability across architectures.”
>
> **Answer 4:**
> Thank you for the suggestion. In this work, we focus on Qwen2.5-VL as the backbone because all related “thinking with images’’ baselines used for comparison—GRIT, DeepEyes, and Visual-SR1—are implemented on top of the same Qwen-based family of models. Using a consistent backbone ensures that the comparisons are controlled and reflect differences in training methodology rather than architectural capacity. Extending DeFacto to other architectures such as InternVL3.5 is feasible and is an interesting direction for future work, but we believe the current evaluation already provides a fair assessment within this research line.

---

> ### Author Response · Authors · 2025-11-19
>
> **Weakness 4 (Minor):**
> “The paper lacks qualitative comparisons on the reasoning processes between DeFacto and GRPO.”
>
> **Answer 4:**
> Thank you for the suggestion. To address this, we added a new section in the revised manuscript Appendix D where we present qualitative comparisons of the reasoning traces produced by DeFacto and GRPO.
>
>
> **Question 1:**
> “In Table 1, the term ‘CF reward’ is not clearly defined. Based on my understanding, CF reward refers to your added components (the region selection coherence and answer correctness rewards) combined with the original GRPO training. Please confirm if this interpretation is correct.”
>
> **Answer 1:**
> Yes, your understanding is correct.  We have clarified this definition more explicitly in the revised manuscript to avoid ambiguity.

---

> ### Author Response · Authors · 2025-11-19
>
> **Weakness 4 (Minor):**
> “The paper lacks qualitative comparisons on the reasoning processes between DeFacto and GRPO.”
>
> **Answer 4:**
> Thank you for the suggestion. To address this, we added a new section in the revised manuscript Appendix D where we present qualitative comparisons of the reasoning traces produced by DeFacto and GRPO.
>
>
> **Question 1:**
> “In Table 1, the term ‘CF reward’ is not clearly defined. Based on my understanding, CF reward refers to your added components (the region selection coherence and answer correctness rewards) combined with the original GRPO training. Please confirm if this interpretation is correct.”
>
> **Answer 1:**
> Yes, your understanding is correct.  We have clarified this definition more explicitly in the revised manuscript to avoid ambiguity.

---

> ### Comment · Reviewer_oCdo · 2025-11-21
> **Official Comment by Reviewer oCdo**
>
> Thank you for the rebuttal. However, I have remaining concerns regarding the inconsistencies between the rebuttal and the main paper:
>
> 1. **Contradiction with Fig. 1**: The rebuttal in W2 ( `GRIT often enters repetitive rethink loops and generates unconstrained bounding boxes, leading to correct answers but unfaithful reasoning. DeepEyes repeatedly triggers zoom-based tool calls and produces many redundant or noisy boxes, resulting in very low IoU scores.` ) is the exact inverse of the behavior presented in Fig. 1, which states that GRIT yields incorrect answers with unfaithful reasoning, whereas DeepEyes succeeds despite noisy boxes.
>
> 2. **Quantitative results refute claims in main paper**: The provided quantitative analysis shows GRIT achieves high accuracy with poor visual grounding (mAP, AP50, and AP75), while DeepEyes performs poorly on both metrics. These results directly undermine the paper's claim that GRIT fails due to unfaithful reasoning and that DeepEyes succeeds despite noisy reasoning.

---

> ### Author Response · Authors · 2025-11-21
>
> Thank you for the follow-up and for pointing out these concerns. We clarify both points below.
>
> 1. **Regarding the relationship to Fig. 1:**
>    Fig. 1 presents representative qualitative cases—GRIT failing due to mislocalized or unfaithful reasoning, and DeepEyes occasionally succeeding despite noisy evidence regions. These observations remain accurate. The additional behaviors described in W2 are not corrections of Fig. 1 but rather *further error patterns* revealed by the larger 1k-image validation set. For example, GRIT sometimes answers correctly due to language priors, and DeepEyes often produces redundant regions that depress IoU scores. These are complementary failure modes that extend, rather than contradict, the cases illustrated in Fig. 1.
>
> 2. **Regarding the quantitative results and the phrasing in the main paper:**
>    We appreciate the opportunity to clarify this point. The main-paper statements such as “GRIT fails due to unfaithful reasoning’’ were intended to highlight the observed mismatch between GRIT’s selected regions and its produced answers, not to imply a single uniform failure mechanism across all samples. The 1k-image evaluation shows that even when GRIT answers correctly, the predicted evidence regions rarely align with the annotated ones, reinforcing the concern about unfaithful reasoning. Similarly, the qualitative statement that “DeepEyes succeeds despite noisy reasoning’’ reflects the fact that, in cases where DeepEyes answers correctly, its grounding quality remains significantly lower than its answer accuracy. The quantitative results therefore refine and support the original descriptions rather than contradict them.
>
> We will update the manuscript text to make these points clearer and to avoid potential ambiguity in interpretation.

---

> > ### Comment · Reviewer_oCdo · 2025-11-21
> > **Official Comment by Reviewer oCdo**
> >
> > I have adjusted the final rating to 4 due to the flawed motivation.
> >
> > Initially, I valued the paper's insight on how noisy thinking affects performance. However, the rebuttal data contradicts this core premise: GRIT achieves high accuracy despite unfaithful reasoning, while the better-grounded GRPO performs worse. This inconsistency invalidates the central hypothesis. Since baselines effectively succeed without faithful reasoning, attributing the proposed method’s performance gains solely to improved grounding becomes unconvincing. Besides, Fig. 1 (which sets the motivation for the paper) is also misleading. Despite the appreciated reward design, significant revision is required to align the claims with the experiment support.

---

> > > ### Author Response · Authors · 2025-11-21
> > >
> > > I would like to reiterate that the motivation of our work is to improve the consistency between reasoning and answers. GRIT obtaining high accuracy but very low faithfulness indicates that it has learned a shortcut by focusing only on the final answer instead of using evidence, which is not a desirable behavior. Qwen2.5-VL plus GRPO shows the opposite imbalance where grounding improves but the predicted answers are often incorrect, meaning the evidence and the final output remain disconnected. DeepEyes exhibits similar inconsistencies. Our method improves both accuracy and faithfulness because the training includes correctness rewards, counterfactual rewards and missing-evidence constraints, which together prevent such shortcuts and encourage the model to keep its reasoning aligned with the visual evidence.

---

> ### Author Response · Authors · 2025-11-22
>
> Thank you for your positive assessment of our work and for pointing out some weaknesses. We have updated the manuscript to address  concerns.
>
> **Weakness 1 (Major) :**
> “The main concern is that the paper does not directly verify whether DeFacto actually achieves its goal of enabling the model’s reasoning to be faithful to the image. Although Table 3 includes an ablation between DeFacto and GRPO, it only compares answer accuracy and does not evaluate whether the reasoning process itself becomes more visually faithful. There should be quantitative results showing whether the thinking process correctly utilizes the predicted visual information after applying DeFacto.”
>
> **Answer 1:**
> Thank you for pointing this out. To directly evaluate whether DeFacto improves the visual faithfulness of the reasoning process, we constructed a new 1k manually annotated validation set. This set was specifically designed to assess whether the model not only answers correctly but also grounds its reasoning in the correct visual regions. For each sample, human annotators label the evidence regions associated with the answer. During evaluation, the model’s predicted answer is treated as the category for computing mAP, AP50, and AP75: we compare the predicted evidence regions for that answer with the human-annotated evidence of the same answer category. This directly measures whether the model’s reasoning is consistent with the visual evidence required for its predicted answer.
>
> The results are shown in Table A.
>
> **Table A: Reasoning Faithfulness on the 1k Human-Annotated Validation Set**
>
> | Model             | mAP  | AP50 | AP75 | Accuracy (%) |
> | ----------------- | ---- | ---- | ---- | ------------ |
> | GRIT              | 0.0  | 0.0  | 0.0  | 73.7         |
> | DeepEyes          | 0.0  | 2.2  | 0.9  | 44.0         |
> | Qwen2.5-VL + GRPO | 20.4 | 28.8 | 18.9 | 65.0         |
> | **DeFacto (ours)** | **30.6** | **36.1** | **24.8** | **79.4** |
>
> These results provide several insights:
>
> 1. GRIT tends to rely on shortcuts and biases, leading to reasonable accuracy but nearly zero grounding alignment. The model often answers without referencing the correct evidence, revealing that the reasoning chain is not visually grounded.
> 2.DeepEyes  improves cropping behaviors but does not reliably identify which regions matter. It often zooms into irrelevant areas, so the reasoning remains weakly grounded despite the additional visual actions.
> 3.GRPO offers some improvements but still lacks explicit constraints that tie reasoning steps to the underlying image evidence, resulting in moderate but inconsistent grounding quality.
> 4.DeFacto achieves the strongest grounding fidelity. Its counterfactual training encourages the model to rely on the evidence that is actually present and to avoid unsupported reasoning. This results in reasoning traces that are more aligned with human-annotated evidence and more consistent with the intended learning objective.
>
> All new evaluations and analyses have been added to Section 4.2 (“Faithful reasoning evaluation”).
>
>
> **Weakness 2 (Major) :**
> “The paper also does not analyze how often other methods fail due to unfaithful reasoning or succeed despite unfaithful reasoning. Showing these ratios before and after using DeFacto would strongly support the claimed motivation.”
>
> **Answer 2:**
> Thank you for the suggestion. We have added a short analysis in Section 4.2 based on the new 1k human-annotated faithfulness set. The grounding metrics in Table A directly reveal how often each method’s predicted evidence fails to match the human-annotated regions. GRIT and DeepEyes show almost zero mAP/AP scores, indicating that even when they answer correctly, their selected regions rarely correspond to the true evidence. GRPO performs better but still shows substantial misalignment between predicted and annotated evidence. In contrast, DeFacto achieves markedly higher mAP, AP50, and AP75, demonstrating that it substantially reduces these unfaithful-reasoning cases. This comparison provides a clear quantification of how DeFacto improves the consistency between reasoning and visual evidence.
>
>
>
> **Weakness 3 (Minor) :**
> “It would be beneficial to evaluate DeFacto on additional model backbones (e.g., InternVL3.5) to assess its generalizability across architectures.”
>
> **Answer 4:**
> Thank you for the suggestion. In this work, we focus on Qwen2.5-VL as the backbone because all related “thinking with images’’ baselines used for comparison—GRIT, DeepEyes, and Visual-SR1—are implemented on top of the same Qwen-based family of models. Using a consistent backbone ensures that the comparisons are controlled and reflect differences in training methodology rather than architectural capacity. Extending DeFacto to other architectures such as InternVL3.5 is feasible and is an interesting direction for future work, but we believe the current evaluation already provides a fair assessment within this research line.

---

> ### Author Response · Authors · 2025-11-22
>
> **Weakness 4 (Minor):**
> “The paper lacks qualitative comparisons on the reasoning processes between DeFacto and GRPO.”
>
> **Answer 4:**
> Thank you for the suggestion. To address this, we added a new section in the revised manuscript Appendix D where we present qualitative comparisons of the reasoning traces produced by DeFacto and GRPO.
>
>
> **Question 1:**
> “In Table 1, the term ‘CF reward’ is not clearly defined. Based on my understanding, CF reward refers to your added components (the region selection coherence and answer correctness rewards) combined with the original GRPO training. Please confirm if this interpretation is correct.”
>
> **Answer 1:**
> Yes, your understanding is correct.  We have clarified this definition more explicitly in the revised manuscript to avoid ambiguity.

---

> ### Author Response · Authors · 2025-11-26
>
> Thank you for your feedback. To address your new concerns, we added a new experiment to further validate our motivation, which emphasizes the importance of ensuring consistency between reasoning and answers.
>
> As shown in Table D, DeFacto outperforms all other variants in both grounding quality and accuracy, confirming that our approach improves both aspects simultaneously.
>
> **Table D: Faithfulness Evaluation**
>
>
> | Model Variant                    | mAP  | AP50 | AP75 | Accuracy (%) |
> | -------------------------------- | ---- | ---- | ---- | ------------ |
> | Qwen2.5-VL (Base)                | 2.3 | 1.2 | 1.5 | 55.1         |
> | Qwen2.5-VL + SFT (no CF)         | 15.7 | 13.9 | 12.8 | 61.4         |
> | Qwen2.5-VL + SFT (CF alignment)  | 18.9 | 16.4 | 15.7 | 63.8        |
> | Qwen2.5-VL + GRPO (no CF reward) | 20.4 | 28.8 | 18.9 | 65.0         |
> | **DeFacto (ours)**|**30.6**|**36.1**|**24.8**|**79.4**|
>
> This new experiment further supports the claim that joint reasoning and answer consistency are key to improving both accuracy and grounding. We will revise the manuscript to ensure this connection is clear.

---

### Author Response · Authors · 2025-11-29
**General response**

Dear AC,
Thank you for taking over the handling of this submission. We would like to briefly summarize the revisions made in response to the concerns raised by reviewers oCdo, 8FmZ, o15t, and JgWv.

We have substantially updated the manuscript, and the main changes are as follows:

1. adding a manually annotated 1,000-image validation set to directly assess evidence localization and reasoning–answer consistency (Section 4.2);
2. expanding comparisons to recent “thinking with images” baselines, including Chain-of-Focus and Pixel-Reasoner (Appendix A);
3. adding evaluation on the MSTI 2.0 dataset covering detection, entity recognition, and answer generation;
4. including additional case studies, error analysis, and visualizations (Section 5 and Appendix D);
5. clarifying the motivation and refining several passages to reduce potential ambiguity.


To address the additional concerns raised during the discussion phase, we performed a new controlled experiment where all variants share the same backbone, data, and hyperparameters. The only difference is whether counterfactual supervision is used. The results (Table F in the rebuttal to JgWv) show that DeFacto achieves higher mAP, AP50, AP75, and accuracy than all alternatives, demonstrating that counterfactual supervision improves both evidence grounding and answer reliability together. This experiment directly addresses the reviewers’ questions about the relationship between grounding and accuracy and confirms that our method strengthens the consistency between reasoning and final predictions.

All updates have been incorporated into the revised manuscript, and a marked-up version in the supplementary materials highlights every change.

Thank you for your consideration.
Best

---

### Note · Authors · 2026-01-27

I have read and agree with the venue's withdrawal policy on behalf of myself and my co-authors.

---

### Meta-Review · Area_Chair_bQgq · 2026-01-05

**Summary:**

The main strengths are (1) the research problem of spurious correctness and mislocalized reasoning in thinking with images is fundamental, (2) the proposed approach with positive, counterfactual, and random-masking paradigms is reasonable and effective, (3) the contributions of the dataset and pipeline are acknowledged, (4) the improvements over strong baselines across multiple VQA benchmarks are consistent. The initial major weaknesses and concerns were (1) the lack of direct evidences on whether grounding quality affects answer accuracy, (2) the proposed approach depends heavily on open-vocabulary detectors and region proposal models, and the noise introduced by these modules may impact the performance, (3) insufficient evaluation on other tasks such as MSTI, which has been provided in the rebuttal.

The initial ratings were 4, 4, 6, 8. After rebuttal, two reviewers who gave positive ratings lowered their scores from 6 to 4 and from 8 to 6. The remaining major concern is the motivation contradicts with new evidence as better grounding ability is not necessary for higher accuracy. The authors argued that the goal is to enforcing consistency between reasoning and visual evidence. However, the inconsistency between grounding ability and task accuracy is not well explained or deeply analyzed, which makes the contributions of the paper less solid. Therefore, the paper requires major revision and clearer explanations. The area chair acknowledged the importance and insights of this work, but encourages authors to further polish the paper and include deeper and consistent analysis in the revision.

**Reviewer Concerns:**

Some of the concerns are addressed with additional evaluation results and clarifications. The remaining major concern is the motivation contradicts with new evidence as better grounding ability is not necessary for higher accuracy. The authors argued that the goal is to enforcing consistency between reasoning and visual evidence. However, the inconsistency between grounding ability and task accuracy is not well explained or deeply analyzed, which makes the contributions of the paper less solid.

**Reviewer Scores:**

After the rebuttal, two reviewers who gave positive ratings lowered their scores from 6 to 4 and from 8 to 6. According to the remaining major concern, the overall ratings would be negative.

---

### Decision · Program_Chairs · 2026-01-26

Reject